# High resolution spatial profiling of kidney injury and repair using RNA hybridization-based in situ sequencing

Haojia Wu [1], Eryn E. Dixon [1], Qiao Xuanyuan[1], Juanru Guo[1], Yasuhiro Yoshimura [1], Chitnis Debashish[2], Anezka Niesnerova[2], Hao Xu[2,4], Morgane Rouault[2] & Benjamin D. Humphreys [1,3] ✉

Emerging spatially resolved transcriptomics technologies allow for the measurement of gene expression in situ at cellular resolution. We apply direct RNA hybridization-based in situ sequencing (dRNA HybISS, Cartana part of 10xGenomics) to compare male and female healthy mouse kidneys and the male kidney injury and repair timecourse. A pre-selected panel of 200 genes is used to identify cell state dynamics patterns during injury and repair. We develop a new computational pipeline, CellScopes, for the rapid analysis, multi-omic integration and visualization of spatially resolved transcriptomic datasets. The resulting dataset allows us to resolve 13 kidney cell types within distinct kidney niches, dynamic alterations in cell state over the course of injury and repair and cell-cell interactions between leukocytes and kidney parenchyma. At late timepoints after injury, C3+ leukocytes are enriched near pro-inflammatory, failed-repair proximal tubule cells. Integration of snRNA-seq dataset from the same injury and repair samples also allows us to impute the spatial localization of genes not directly measured by dRNA HybISS.

The kidney is a highly complex organ, consisting of hundreds of thousands of nephrons supported by a rich peritubular vascular network as well as interstitial fibroblasts, macrophages, lymphocytes and perivascular cells. These disparate cell types play critical roles in maintaining homeostasis as well as responding to kidney injury. Recently several single-cell multi-omic analyses have greatly advanced our understanding of kidney cell states both in healthy and diseased kidneys[1–6]. A limitation of these single-cell multi-omic analyses is that they require cell or nuclear dissociation as a first step, resulting in the loss of spatial information which precludes a full analysis of how cells interact with each other in their tissue microenvironment[7].

Recent advances in spatial transcriptomic technologies have begun to allow high-throughput quantification of RNAs within an intact tissue section. We have previously applied a next-generation sequencing (NGS)-based spatially resolved transcriptomic (SrT) approach, Visium by 10X Genomics, to map transcriptional changes during the full acute kidney injury timecourse[8]. This allowed us to reconstruct the spatial interactions between injured proximal tubular cells (InjPT) and macrophages[8]. Other groups have used the same technique to localize macrophage subtypes to their different kidney regions and map changes in their spatial localization during AKI[9]. Slide-seq is another NGS-based SrT implementation which has been used to analyze transcriptomic changes in glomeruli during diabetic kidney disease, revealing a unique glomerulus-associated TREM2+ macrophage population[10]. Despite the power of these NGS-SrT techniques for revealing the spatial relationships between kidney cell types, they are all limited by relatively low resolution. For example, Visium has a spot size of 55 μm and therefore cannot resolve individual cell types within a glomerulus which itself is ~50–100 μm. Slide-seq V2 has improved spatial resolution of 10 μm but is limited by substantially

[1]Division of Nephrology, Department of Medicine, Washington University in St. Louis School of Medicine, St. Louis, MO, USA. [2]10X Genomics, Pleasanton, CA, USA. [3]Department of Developmental Biology, Washington University in St. Louis School of Medicine, St. Louis, MO, USA. [4]Present address: Aplex Bio AB, Solna, Sweden. ✉e-mail: humphreysbd@wustl.edu

lower gene detection sensitivity and a smaller capture area (3 mm in diameter).

Here we aimed to generate a high-resolution transcriptomic map of kidney cell types during injury and repair using in situ sequencing (dRNA HybISS) by Cartana (10X Genomics)[11,12]. We selected 50 custom kidney gene probes and added 150 pre-designed probes for immune and lung cell types. Out of this 50 gene custom kidney panel, we successfully detected expression of 37 of these genes (74%). In total we generated seven SrT datasets: one Visium dataset to compare performance of NGS-SrT and ISS-SrT on the same sample, and six ISS datasets to study differences in spatial gene expression patterns between healthy male and female kidneys or during the male murine ischemia-reperfusion (IRI) injury timecourse. We developed a new computational pipeline which we call CellScopes, for the rapid processing, visualization and mulit-omic integration of this high dimensional spatial data. This allowed us to resolve nearly all major kidney cell types from murine cortex, medulla and papilla and provided us a high-resolution view of kidney niches including the glomerulus, the interstitium and renal artery. We could observe dynamic cell state changes during IRI, both in cell proportions and spatial location during the injury and recovery phases. This single-molecule level spatial technique also allowed us to dissect the heterogeneity and spatial distribution of the immune subtypes in the recovery phase of AKI, and to identify the interactions between the failed-repair proximal tubule and the inflammatory macrophages. To extend the power of the dataset, we integrated the dRNA HybISS data with 10X Visium for directly decomposing the cell-type proportion in each Visium spot. This provided a ground truth for testing the accuracy of various cell-type deconvolution tools that rely on computational prediction. Finally, we integrated a single nucleus RNA-sequencing (snRNA-seq) onto our high-resolution SrT dataset which enabled us to impute gene expression patterns for genes that were not measured by in situ sequencing. This allowed us to validate the spatial expression patterns of the sexually dimorphic genes and disease genes in IRI. This kidney injury atlas serves as a high-resolution resource to better understand AKI, and the underlying tool developed is broadly applicable for the analysis of high-resolution SrT datasets across tissues.

## Results

### Optimized workflow for dRNA HybISS and its application in kidney research

The application of ISS-based SrT has primarily been applied in neuroscience and developmental biology[13–16]. To evaluate the utility of this approach to study kidney injury, we applied a probe-set targeting 200 mouse genes during five different timepoints in an ischemia-reperfusion injury timecourse spanning the acute injury phase (hour 4, hour 12, and day 2) and the repair phase (week 6) (Fig. 1A). We chose these timepoints in part because we have previously profiled this same IRI timecourse by snRNA-seq[17] which would allow for downstream integration of these datasets. Kidney injury and repair-specific genes were selected based on our prior snRNA-seq analysis highlighting specific injury and repair cell states. In addition, two immune cell gene panels were selected to elucidate interactions between immune cells and kidney parenchyma. To study sexual dimorphism, we incorporated a female sham sample for dRNA HybISS to compare with the male sham sample (Fig. 1A). Finally, we performed Visium and dRNA HybISS on the same healthy male sample to illustrate how integrating whole transcriptome and targeted in situ imaging data offers complementary and enriching biological insights (Fig. 1A). Sample preparation and dRNA HybISS protocols were performed according to manufacturer's instructions and all timepoints were processed in the same run to minimize batch effects (Fig. 1A).

We compared different cell segmentation algorithms based on DAPI staining of nuclei. First we utilized the Watershed algorithm which has been used in the literature to split clustered objects[18,19]. This resulted both in missed nuclei as well as clusters of multiple nuclei that were incorrectly designated as a single segment (Supplementary Fig. 1A), likely due to the dense packing of cells in the kidney which decreases the accuracy of the watershed algorithm[20]. Next, we segmented the nuclei using CellPose, a machine learning based segmentation algorithm[21]. This generated superior results with few missed nuclei and many fewer multiple nuclei clusters (Supplementary Fig. 1A). After using Cellpose to segment nuclei, we added cell boundaries and assigned each detected transcript to individual cells using the Baysor algorithm[22]. This algorithm considers both the RNA transcriptional composition and cell morphology, modeling each cell as a combination of its spatial location and gene identity of each molecule, and has been shown to be superior to other segmentation tools in terms of achieving higher segmentation accuracy, detecting more cells and providing improved molecular resolution[22]. This approach successfully assigned specific transcripts to individual kidney cells (Supplementary Fig. 1B). Cells survived from the Baysor cell filtering and greater than 5 transcripts detected were selected for downstream analysis. Cell counts for each sample were summarized in Supplementary Table 1.

### A toolkit, CellScopes, for highly efficient spatial data analysis

Existing tools[23–25] for spatial data analysis are designed to process input data from particular spatial transcriptomics techniques integrated within their systems such as Visium or Xenium. New spatial data types, such as the dRNA HybISS data introduced in this study, either couldn't be processed by these tools or required significant effort to adapt the format to meet their input specifications. In addition, these tools fall short of visualizing the complex structure in the kidney and fail to adequately illustrate the dynamic changes in gene and cell spatial relationships. To overcome these challenges, we developed a new software package, CellScopes.jl, to perform downstream data analysis and visualization after cell segmentation (Fig. 1B), with functionalities specifically tailored for kidney spatial data analysis (See Methods). This implementation is open-source and written in Julia, a programming language known for its high-performance capabilities and efficient execution[26]. CellScopes can process a diverse range of modalities, including datasets from single-cell profiling (scRNA-seq and scATAC-seq), sequencing-based SrT profiling (10X Visium and Slide-seq), and imaging-based SrT profiling (dRNA HybISS as presented in this study, as well as MERFISH, SeqFish, STARmap, and 10X Xenium) (Fig.1B). CellScopes is publicly available on GitHub (https://github.com/TheHumphreysLab/CellScopes.jl).

To highlight the distinctive features of CellScopes, we analyzed two public human kidney datasets from Xenium and Visium using CellScopes, Seurat V5[27], Giotto[24], and Squidpy[25] (Supplementary Figs. 2–6). For imaging-based data analysis, we clustered the cells from a human kidney Xenium dataset with CellScopes and identified the cell types based on the marker gene expression and their spatial distribution patterns (Supplementary Fig. 2A, B). For direct comparison, we applied the same cell annotations to the clusters identified from other tools and focus on the same field of view (FOV) to highlight the visualization features. CellScopes facilitates easy selection and zooming into specific FOV to examine intricate kidney structures by adding a grid system (Supplementary Fig. 2C), a direct visualization of transcripts, genes and cell-type annotation in cells as data point or cell polygons format (Supplementary Fig. 2D–H) and direct imputation of the missing genes by integrating the popular gene imputation tools such as SpaGE[28], gimVI[29] and tangram[30] (Supplementary Fig. 2I). Some of these important features are missing in other tools such as Seurat V5 (Supplementary Fig. 3), Giotto (Supplementary Fig. 4), Squidpy (Supplementary Fig. 5). For sequencing-based SrT data analysis, CellScopes.jl also can incorporate any high-resolution images as the background layer that allows colocalizing the cell-type annotations or

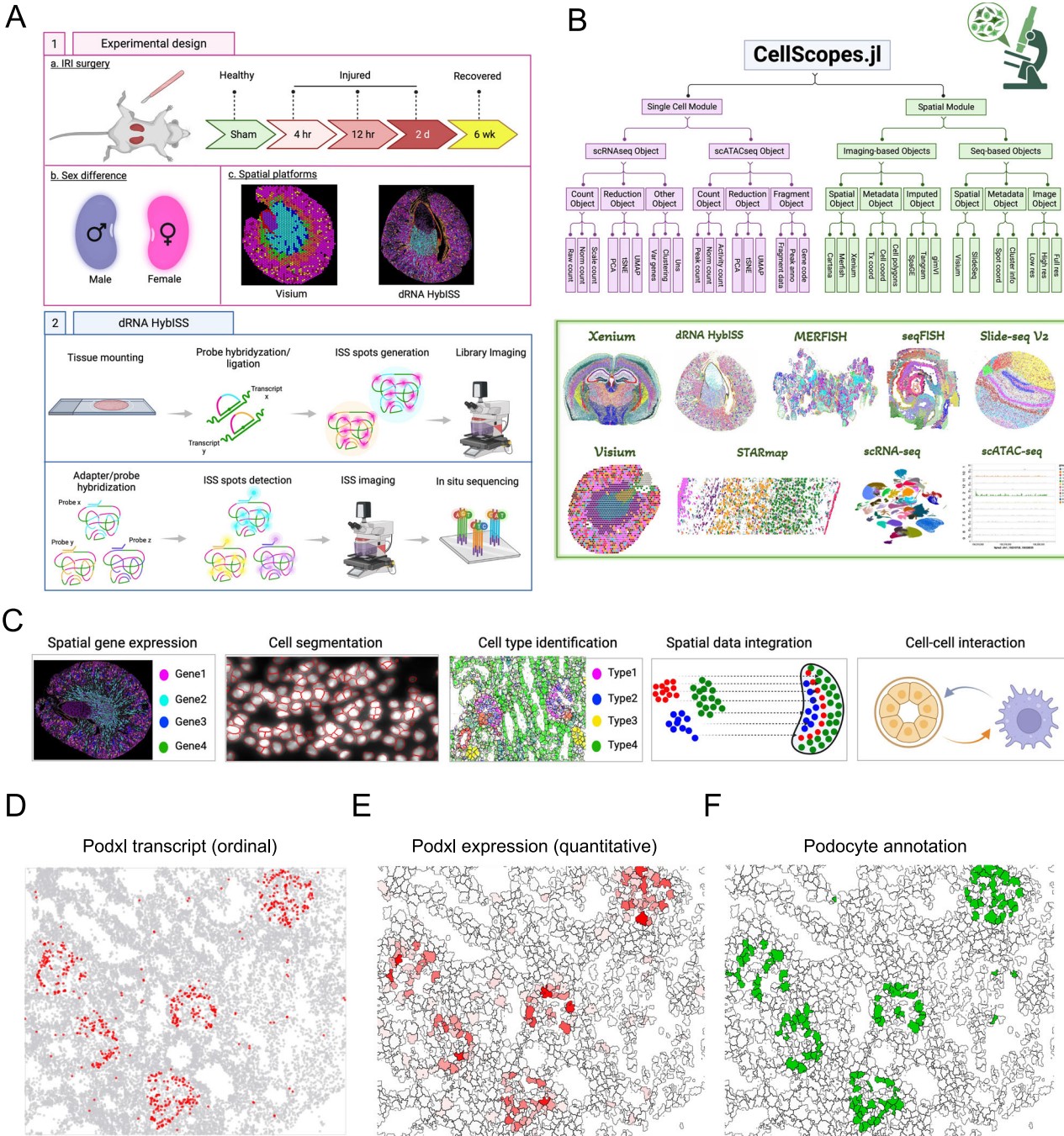

Fig. 1 | **dRNA HybISS workflow, experimental design and computational analysis. A** Schematic of using dRNA HybISS to study AKI. Schematic was created with BioRender. **B** A Julia package, CellScopes.jl, was developed for spatial data processing, analysis and visualization. Image was created with BioRender. **C** Outline of the spatial analysis pipeline used in this study, created with BioRender. **D**–**F** CellScopes.jl allows for visualization of gene expression on cells as data points (**D**), segmented polygons (**E**), and cell-type annotations (**F**).

gene expression on top of precise histologic features (Supplementary Fig. 6). This can significantly facilitate the interpretation of the cell-type annotation and gene expression pattern by showing the delicate kidney structure (such as the glomerulus) in histological staining images (Supplementary Fig. 6B). Other tools only allow importing the images from the 10X SpaceRanger output, which is not high resolution enough to visualize the kidney structure (Supplementary Fig. 6B). The primary advantages of CellScopes are summarized in Supplementary Table 2.

We then used CellScopes to analyze the dRNA HybISS data. Data processing for dRNA HybISS with CellScopes consists of the following steps (Supplementary Fig. 1C): (1) Input the spatial coordinate files and the gene-by-cell count matrix from cell segmentation. (2) These datasets are stored in a data structure called CartanaObject. (3) Analysis is performed directly on the CartanaObject (for example cell clustering, cell–cell distance computation, coordinate transformation, scRNA-seq integration) and the output is saved in the same object. (4) Finally, a variety of different functions are then used to visualize the analyses. We used CellScopes to complete the primary analyses in this study including gene and cell visualization after segmentation, data integration, cell proximity analysis (Fig. 1C). Transcripts, genes and cell types in complicated kidney structures (such as glomeruli) can be robustly revealed by CellScopes (Fig. 1D–F).

## Spatial cell-type classification of the mouse kidney

We categorized the cell types in each dataset by using the transcript clustering results from the Baysor segmentation[22]. Baysor first employs a Markov Random Fields (MRFs) approach to cluster the transcripts. It then assigns a cluster label to each segmented cell, based on the highest probability of a particular cluster label's occurrence within the cell, and designating the most frequently occurring label to that cell. We selected a day 2 IRI kidney as a representative to demonstrate the resolution of dRNA HybISS in classifying cell type, given that kidney at this timepoint contains kidney cell types in both healthy and disease states. The cell annotations based on the cluster labels assigned by Baysor are very consistent with the cell annotations derived from Seurat clustering (Supplementary Fig. 7). Overall, the clustering algorithm identified 13 molecularly distinct cell populations in the kidney (Fig. 2A). This includes most major kidney cell types previously detected in scRNA-seq studies[17,31,32] including rare cell types such as podocyte, juxtaglomerular apparatus (JGA) and glomerular endothelial cells (gEC) (Fig. 2A). To determine how many kidney cell types from the published scRNA-seq datasets can be identified in our dRNA HybISS dataset, we performed label transfer using Seurat to transfer the cell-type annotations from two separate mouse kidney scRNA-seq datasets[17,33] onto our spatial data in the sham kidney. This analysis confirmed that the majority of kidney cell types can be accurately mapped to clusters in our spatial data (Supplementary Fig. 8). However, due to the constraint on the number of genes included, cell types lacking marker genes in our probe panel were undetectable in our data. These include the parietal epithelial cells (PEC), the thin limb of the loop of Henle (ATL-DTL), and certain subpopulations within a cell type (e.g., mTAL, cTAL1, and cTAL2 from Kirita et al. dataset[17] were indistinguishable within the TAL cell cluster) (Supplementary Fig. 8). CellScopes.jl allowed for the detection of fine kidney structures from different kidney compartments including the glomerulus, cortex, outer medulla, inner medulla, cortex, and renal arteries, preserving tissue morphology (Fig. 2A). For example, we could precisely identify the spatial relationship of podocytes, Renin+ JGA cells and gEC (Fig. 2A). The unique expression of cell markers defined for each cell type reflects the accuracy of our cell segmentation approach (Fig. 2B). The spatial expression of these anchor genes colocalized with the spatial distribution of predicted cell types (Supplementary Fig. 9). In addition, our high resolution of dRNA HybISS data can resolve individual intercalated interspersed among principal cells in collecting duct (Supplementary Fig. 10A). Finally, a detailed subclustering analysis further identified two distinct subpopulations within the fibroblast cell type (Supplementary Fig. 10B), each with a unique spatial distribution (Supplementary Fig. 10C, D). One subpopulation is characterized by the expression of pericyte/myofibroblast markers such as Tagln and Acta2, while the other is marked by the expression of differentiated fibroblast markers, Col1a1 and Col1a2 (Supplementary Fig. 10E).

Since our SrT dataset generated spatial coordinates for each single cell, we could use these coordinates to validate certain predicted cell-type annotations based on expected proximity. For example, glomerular endothelial cells are distinct from arterial endothelial cell subpopulations based both on their gene expression (Ehd3 for gEC and Eln for arterial EC) but also spatial localization (glomerulus vs. artery). To confirm our cell annotations, we compared the cell–cell distance between podocytes and these two EC subtypes, reasoning that gEC would be much closer to podocytes than arterial EC (see Methods). We designed an algorithm (implemented in CellScopes) to measure the cell density of any cell type within a given radius centered by each podocyte and uses the cell density as a readout to measure the proximity between podocyte and other cell types (Supplementary Fig. 11A). By measuring the cell density of the EC subtypes according to distance from podocytes, we

confirmed that gEC was much closer to podocytes at short distances (2-20 cells) but this distinction is lost at higher cell distances (200–600 cells) (Fig. 2C).

To measure gene expression dynamics and cell-type composition changes across the entire kidney, we created a kidney coordinate system in the CellScopes.jl package (see Methods). In this system, the position of a cell is defined by the kidney depth (measuring the distance of each cell type to the kidney capsule) and the kidney angle (measuring the slope of each cell with respective to the positive x-axis of the new coordinate system) (Supplementary Fig. 11B). With these coordinates, we were able to visualize cell type and transcript distribution changes from outer cortex to papilla (Fig. 2D). As expected, the cell types residing in cortex, such as PT and podocytes, have lower kidney depth values, whereas cell types in the papilla such as principal cells (CD-PC) and urothelial cells (Uro) have higher kidney depth values (Fig. 2D). The distribution of cell-type-specific markers is consistent with the distribution of the cell types (Fig. 2D), further corroborating the accuracy of our cell-type classification.

## Sex-related gene expression diversity in proximal tubule

Sexually dimorphic gene expression has been reported in proximal tubule from recent cell-type-specific bulk RNA-seq and scRNA-seq studies[34,35]. To investigate this further, we next generated in situ sequencing (dRNA HybISS) by Cartana (part of 10X Genomics) datasets comparing healthy male and female mouse kidney. Unsupervised clustering identified the same kidney cell populations from both sexes (Fig. 3A). We annotated cell types based on markers identified from a previous scRNA-seq study[35] of sexually dimorphic gene expression (Supplementary Fig. 12A, B). These anchor genes did not exhibit sexually dimorphic gene expression (Fig. 3B, Supplementary Fig. 12C). We selected five marker genes from the scRNA-seq study corresponding to specific cell types in both sexes. These were Inmt (PTS2), Rnf24 (PTS3), Csf1r (PTS3), Scel (ATL), and Msln (Uro) (Supplementary Fig. 12D). We mapped the expression of these genes onto our spatial coordinates and confirmed the expected cell-specific spatial expression pattern in both male and female (Fig. 3C). This result demonstrates that high-resolution spatial transcriptomic data can validate the expected expression patterns of cell-specific marker genes.

We next sought to characterize the expression patterns of sexually dimorphic genes. Although our initial probe set was limited, we wanted to be able to predict gene expression patterns transcriptome wide. We therefore compared methods to impute gene expression. We evaluated the performance of tangram[30], SpaGE[28] and gimVI[29]. SpaGE had superior accuracy in our dataset so this method was selected (Supplementary Fig. 13). We could then validate the expression patterns for sexually dimorphic genes in proximal tubule identified in the scRNA-seq dataset (Fig. 3 and Supplementary Fig. 12E). The spatial distribution of these genes is consistent with the expected proximal expression pattern (Fig. 3E). For example, Abcc3 (an ATP-binding cassette (ABC) transporter) and Akr1c18 (a Aldo-keto reductase) are expressed in the S3 segment of the female PT only, whereas two of the cytochrome P450 superfamily enzymes, Cyp2e1 and Cyp4a12, are expressed in the male PT only (Fig. 3E). Interestingly, Cyp2e1 null mice have been shown to be protective against AKI[36], potentially providing a clue to explain why female mice are resistant to AKI.

## Integration of Visium and dRNA HybISS for cell-type deconvolution

Lower resolution but with genome deep detection capacity, Visium spots can be deconvolved to estimate cell-type proportions on each spot using a wide range of methods[24,37,38]. Importantly, these methods are computational predictions and not direct measurements. We therefore sought to leverage our ISS data to directly decompose cell-type proportions in each Visium spot by comparing a new Visium

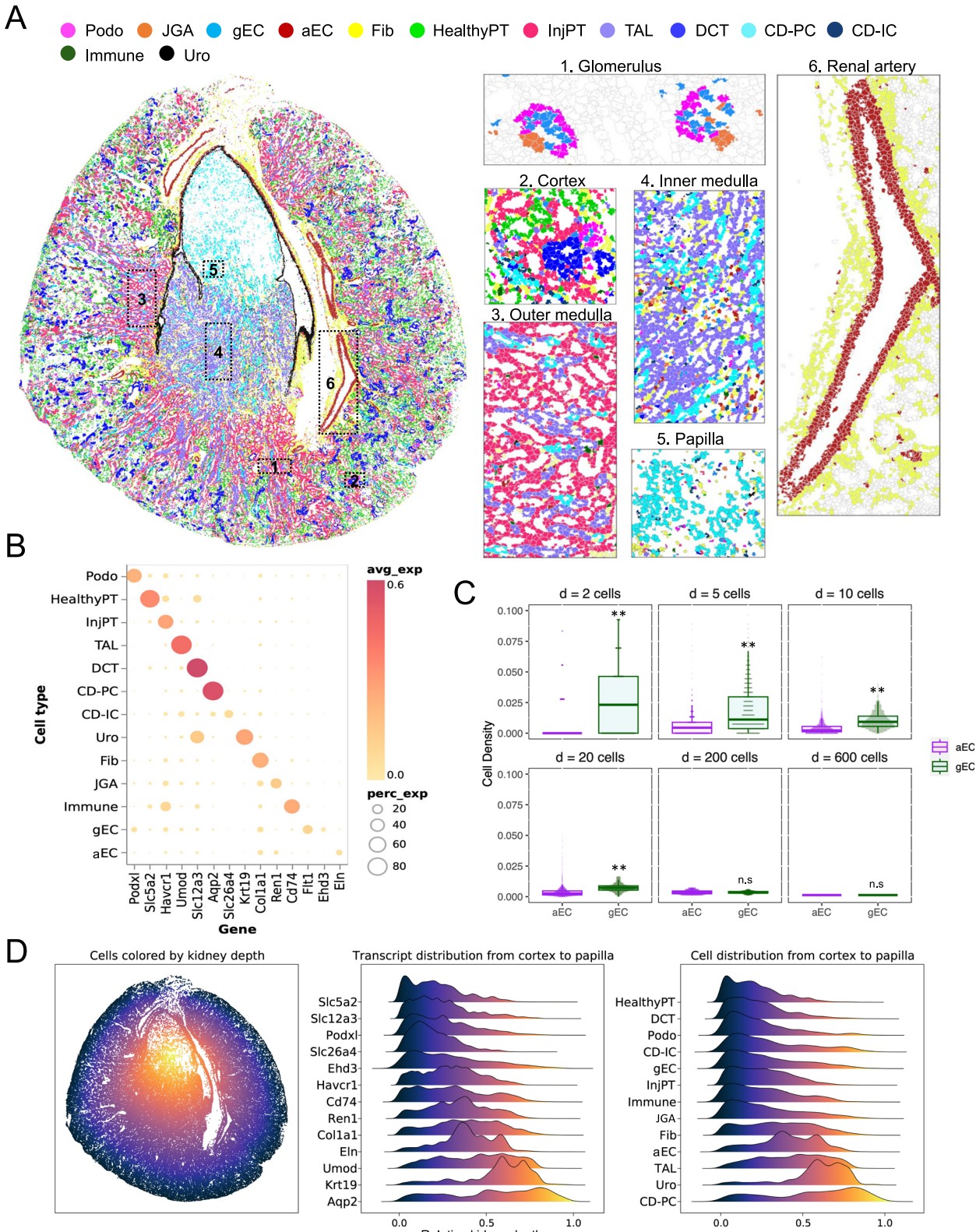

**Fig. 2 | Major cell types and their spatial organizations in the kidney as revealed by dRNA HybISS. A** Spatial distributions of major cell types in the whole kidney and close-ups of cell-type organizations in glomerulus, cortex, outer medulla, inner medulla, papilla and renal artery. Podo podocytes, JGA juxtaglomerular apparatus cells, gEC glomerular endothelial cells, aEC arterial endothelial cells, Fib fibroblasts, HealthyPT healthy proximal tubular cells, InjPT injured proximal tubular cells, TAL thick ascending limb of Loop of Henle, DCT distal deconvoluted tubule, CD-PC collecting duct principal cells, CD-IC collecting duct intercalated cells, Immune immune cells, Uro urothelial cells. **B** Expression known marker genes to define the cell-type identity. **C** Spatial distance between the two EC subtypes and the podocyte. $n = 2812$ podocytes. The line inside the box of the box plot represents the median and the boxes indicate 25th/75th percentile. Solid lines represent the full range of the distribution. Mann–Whitney U test was performed to determine the significance of the difference. **D** Change of transcript and cell proportion from cortex to papilla. **, $P < 0.001$; n.s. not significant.

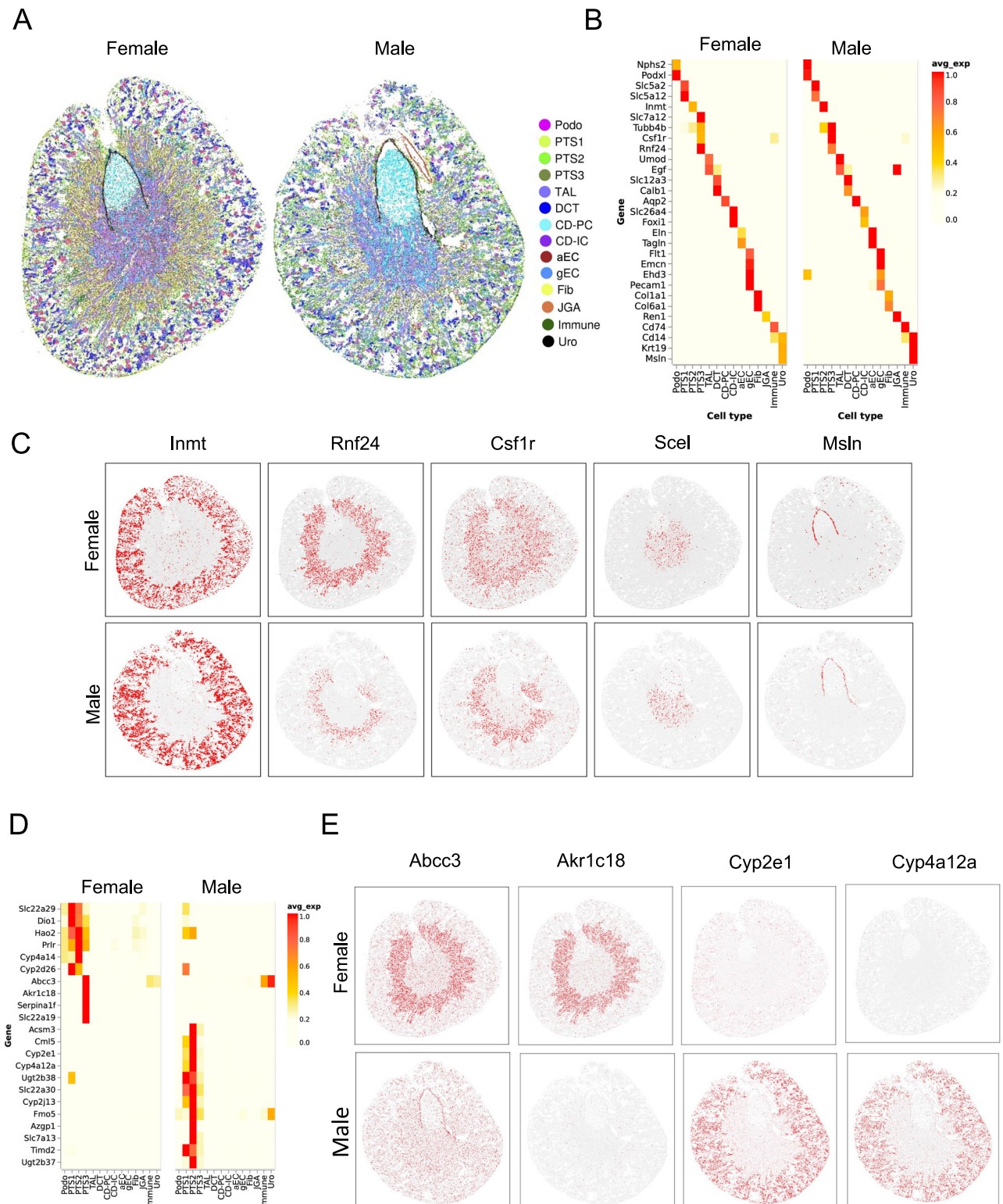

**Fig. 3 | Spatial conserved and divergent gene expression in female and male kidneys. A** Same cell types were identified from female and male kidneys. **B** Female and male kidney cell types express conserved markers. **C** Spatial expression of cell markers were validated by dRNA HybISS. **D** Expression of the sex dimorphic genes in proximal tubule. **E** Spatial distribution of the sex dimorphic genes in the kidney. PTS1 proximal tubule S1 segment, PTS2 proximal tubule S2 segment, PTS3 proximal tubule S3 segment.

dataset from the same kidney that was subjected to imaging-based SrT by dRNA HybISS. With Visium, we were only able to classify 8 kidney cell types based on the mixed gene expression from each spot (Fig. 4A), far fewer than the 14 cell types identified in the corresponding dRNA HybISS dataset (Fig. 3A). The expression of cell-type-specific markers in each cell population identified by Visium was also not as clean as those cell types identified by dRNA HybISS (Fig. 4B), reflecting the higher accuracy of dRNA HybISS for cell-type classification. We then selected and aligned the kidney regions that shared common kidney structures between Visium and dRNA HybISS. After

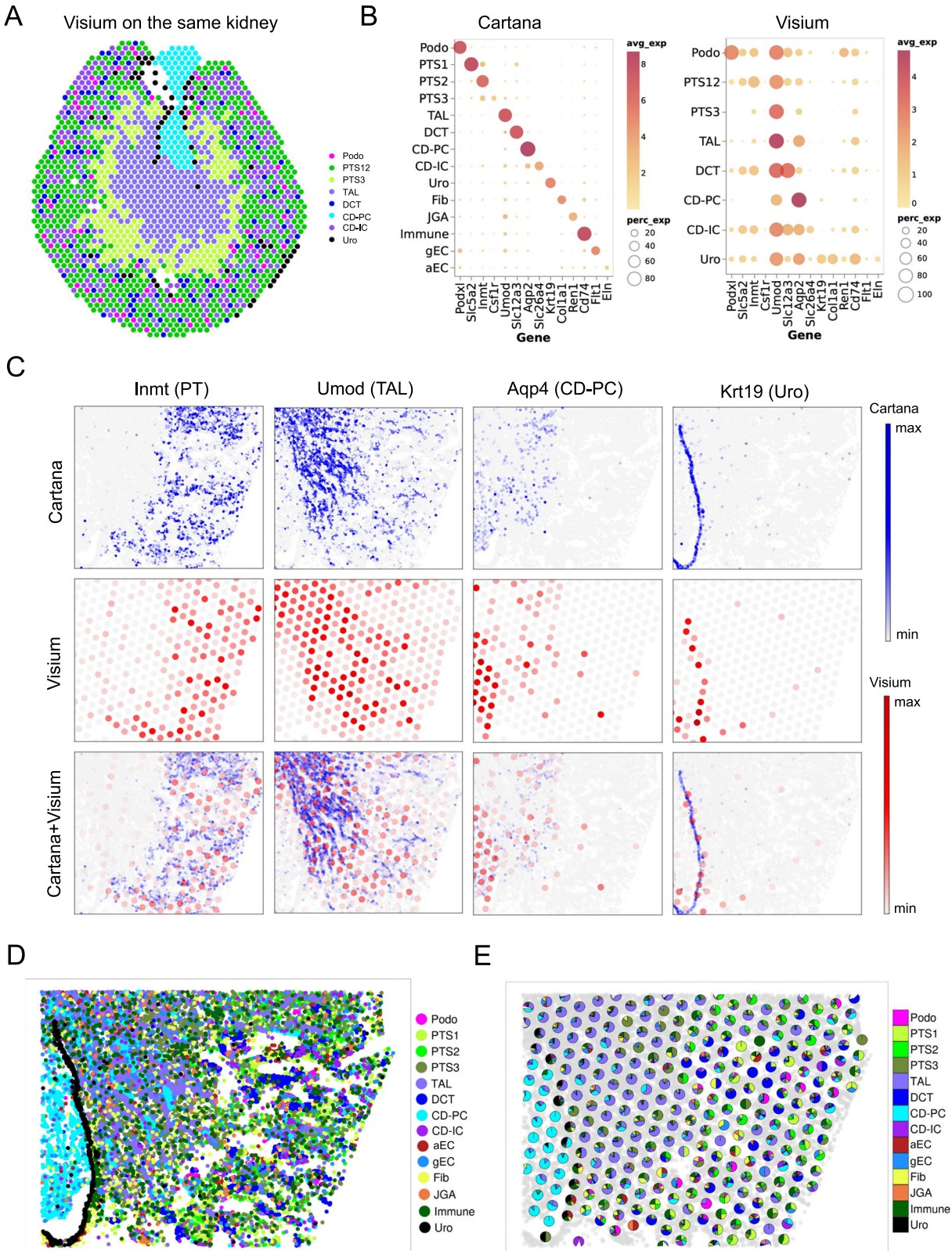

**Fig. 4 | Visium and dRNA HybISS integration to measure the cell-type composition. A** Cell-type classification by Visium on the kidney section adjacent to the section that has been profiled with dRNA HybISS. **B** Dotplot to show the expression of cell-type-specific markers in the cell types identified by Visium and dRNA HybISS. **C** Overlay of the Visium and dRNA HybISS images to inspect the cell-type-specific markers expression on each modality. **D** Visualization of the cell types in dRNA HybISS. **E** Using the cell-type information from dRNA HybISS to measure the cell-type composition for each Visium spot.

graph alignment, the spatial expression cell markers for PT (Inmt), TAL (Umod), CD-PC (Aqp2) and Urothelium (Krt19) matched between Visium and dRNA HybISS (Fig. 4C). This accurate graph alignment allowed us to use the cell-type information from dRNA HybISS (Fig. 4D) to directly measure the cell-type composition in the Visium spot (Fig. 4E). Since this cell-type enumeration approach is from direct measurement, it can serve as ground truth to test the robustness of the cell-type deconvolution tools that are based on computational prediction. We therefore performed cell-type deconvoluton on the each Visium spot from the same region using four different tools including Spotlight[38], RCTD[39], TACCO[40] and STdeconvolve[41]. To benchmark the results from different tools, we focused on the papilla region since the cell types in papilla were known to be relatively pure (only PC). In this analysis, TACCO and STdeconvolve demonstrated superior performance in our kidney Visium data because the cell-type proportions that were deconvolved by TACCO and STdeconvolve exhibited higher correlation with those obtained through direct measurement (Supplementary Fig. 14).

### dRNA HybISS detects spatial dynamics of cell-type distribution and gene expression in AKI

Our previous study from scRNA-seq revealed important cell states and cell-type-specific gene expression change during the timecourse of AKI[17]. A central question is how these cell states and disease genes are distributed in the kidney. To answer this question, we performed ISS-based SrT with dRNA HybISS on the full IRI timecourse. We annotated cell types from a combined dataset of all samples after correcting for batch effects using the Seurat integration algorithm. This leads to slight differences in the cell-type labels. It is noteworthy that distinctions arise primarily in the PT: while single-sample clustering can discern different PT subtypes such as PTS1, PTS2 and PTS3 (Fig. 4B), this difference was not captured when clustering the combined dataset because injury causes downregulation of PT segment-defining markers (Fig. 5A and Supplementary Fig. 15). Consistent with our earlier findings from snRNA-seq and lineage tracing studies[42], the number injured PTs increases in the acute phase of IRI and returns to normal after repair (week 6) (Fig. 5A). As expected, the spatial distribution of the injPT cell state and the injury marker Havcr1 expression extends throughout the cortex at early timepoints (Fig. 5 and Supplementary Fig. 16). Based on bulk profiling data, it has been hypothesized that persistent renal parenchymal injury in AKI models drives chronic inflammatory responses and leading to the AKI to CKD transition[43]. We could utilize our high-resolution SrT dataset to visualize increased immune cell infiltration at week 6, despite apparent successful epithelial repair, supporting this hypothesis (Fig. 5A, B, and Supplementary Fig. 16A). Intriguingly, the fibroblast population proliferates in the early phase of AKI (peaking at day 2) (Fig. 5A, B, and Supplementary Fig. 16A), suggesting an early activation of this cell type. This is consistent with the data from another study on the same animal model[44]. From our snRNA-seq data, we identified a number of disease genes that were expressed in timepoint-specific and cell-type-specific expression patterns as Cxcl1, Plin2, and Gsta1 (Supplementary Fig. 16B). dRNA HybISS was able to validate the temporal expression pattern of these genes and also provided the additional information about the spatial distribution of each gene from the cortex to the papilla (Fig. 5C). For example, Glutathione S-Transferase Alpha 1 (Gsta1) is a newly identified disease gene whose expression was activated in PT during the injury phase of AKI. We validated the spatial expression pattern for these genes that were activated at 12 h and 2d of IRI and restricted to cortex (Fig. 5C).

### Gene imputation expands dRNA HybISS measurements of disease genes to genome scale

Given the high accuracy of SpaGE[28] gene imputation in the healthy kidney (Supplementary Fig. 13), we next asked whether gene imputation could allow us to predict the spatial expression for genes whose expression is temporally modulated by injury. We selected the cell-type-specific disease genes using our previous snRNA-seq dataset[17] on the same IRI model (Fig. 6A and Supplementary Fig. 17A). SpaGE accurately predicted the spatial distribution of the disease genes including Nox4 for PT, Tarm1 for TAL, Klhl3 for DCT-CNT, Frmpd4 for CD-PC (Fig. 6B) along with four disease genes for PTS3 including the known disease markers such as Krt20 (Supplementary Fig. 17B). In addition, the expression changes of these genes in injury phase were also accurately predicted by the gene imputation algorithm (Fig. 6B and Supplementary Fig. 17B). Nox4 (NADPH oxidase 4) in relation to acute kidney injury (AKI) appears to have a complex and controversial role. Prior study showed that expression of Nox4 is upregulated in day 1 of IRI but the genetic ablation of this gene exacerbated the acute kidney injury[45]. our snRNA-seq gene measurement and spatial gene imputation both showed a downregulation of Nox4 in the injury phase of IRI (Fig. 6A, B). This finding aligns with published bulk RNA-seq datasets from the same IRI model (Supplementary Fig. 17C, D). By immunofluorescent staining, we validated that Nox4 expression is downregulated at Day 2 of IRI and the downregulation of Nox4 was colocalized to the upregulation of Havcr1 (Fig. 6C).

### Immune cell heterogeneity and cell–cell communications

We have previously documented the existence of a pro-inflammatory and pro-fibrotic cell state that we have termed "failed-repair proximal tubule" and other have called "maladaptive repair[6,17,46]." Since we could observe substantial immune cell infiltration at week 6, and this failed-repair cell state also exists at week 6 (Supplementary Fig. 18A, B), we next asked whether immune cells were enriched near failed-repair proximal tubule. We subclustered immune cells at week 6 using Seurat and annotated cell types based on the unique gene expression in each cluster (Fig. 7A). This analysis identified six distinct immune subtypes, including macrophages, Ly6e+ lymphocytes, T cells, Cxcl10+ macrophages, C3+ monocytes and plasma cells (Fig. 7A, B). It is important to note that our gene panel also included marker genes for neutrophil and other lymphocyte subtypes such as B cells. However, these genes were largely undetected (Supplementary Fig. 18C). Therefore, these immune cell types were not identified in our dataset. We next mapped these immune subtypes to the immune cell types identified from a previous scRNA-seq study[47] (Fig. 7C). This comparative analysis allowed us to refine the annotation of each immune subtype. The spatial distribution patterns for the immune subtypes were distinct from each other (Fig. 7D). For example, macrophages and lymphocytes are distributed throughout the whole kidney, whereas Lys6c2+ monocytes were confined in the cortex (Fig. 7D). Using Visium, we have previously detected changes in leukocyte–epithelial cell interactions during kidney repair at week 6[8]. Due to the low resolution of the Visium data, it remains unknown what immune subtypes directly interact with tubular cells. To reveal the spatial interaction between the immune subtypes and the injured PT, we used a cell-enrichment approach reported by Lu et al.[48] to compute the proximity between each immune subtype and the failed-repair PT. We calculated the probability of cell-type pairs in a neighborhood within a given searching area (radius = 50 μm). We then computed the enrichment of cell-type pairs in spatial proximity after normalized to the control probability based on random pairing. This analysis revealed a close proximity between C3+ monocytes and the FR-PT (Fig. 7E), suggesting that the FR-PTs are recruiting C3+ monocytes in situ. Indeed, when we projected the cells back to the kidney, we observed a close distance in space between these two cell types (Fig. 7F, G).

## Discussion

In this study, we successfully applied dRNA HybISS (Cartana, part of 10X Genomics), to spatially resolve gene expression patterns in

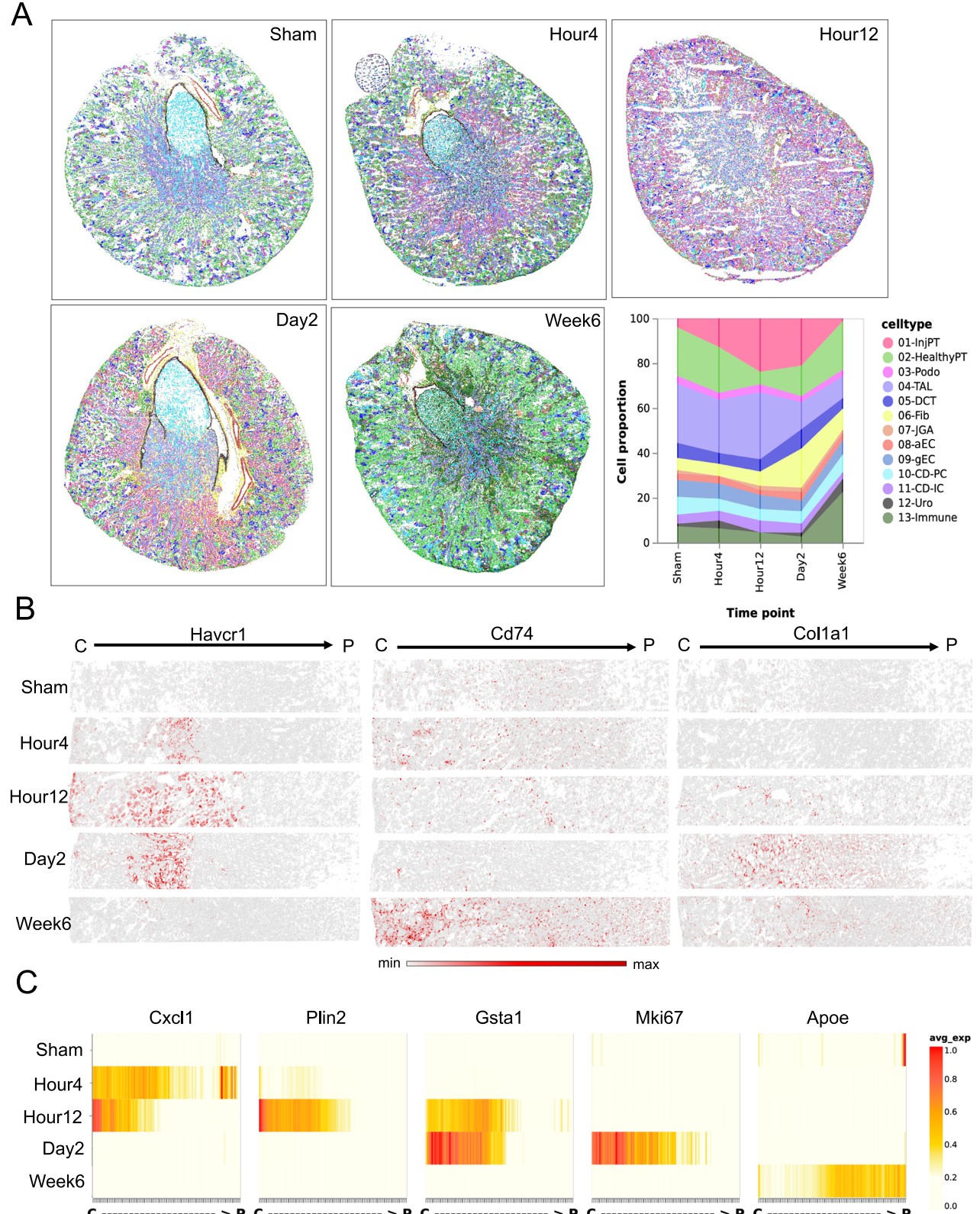

**Fig. 5 | Spatial dynamics of cell type and gene expression changes in acute kidney injury. A** Cell-type composition in IRI timecourse. **B** Spatial expression patterns of the known disease genes that marks injured PT, immune cells and fibroblasts in IRI timecourse. **C** Spatial-temporal expression of disease genes in each phase of AKI. C cortex, P papilla.

murine kidney at cellular resolution. We generated a large dataset of transcriptional changes in cell-type-specific genes between sexes and during kidney injury and repair. We also developed a new tool, CellScopes.jl, to aid in data processing, integration, analysis, visualization, and interpretation. With this tool, we computationally reconstructed the kidney structure cell-by-cell and delineated the cell-type compositional changes along the kidney axis from the outer cortex to the papilla. We validated the sex dimorphic genes

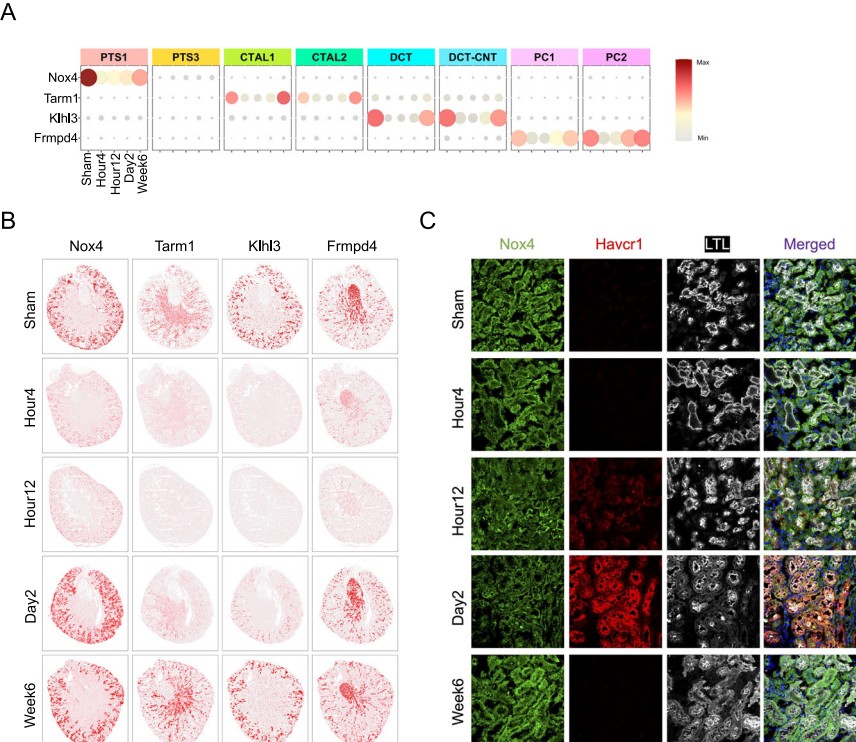

Fig. 6 | Gene imputation for IRI timecourse and independent validation of imputation. A Cell-type-specific expression of the disease genes Nox4, Tarm1, Klhl3, and Frmpd4 across IRI timecourse as revealed by snRNA-seq. B Visualization of imputed gene expression and spatial distribution for Nox4, Tarm1, Klhl3, and

Frmpd4. C Validation of Nox4 expression in kidney tissue across the IRI timecourse. Kidney section was costained with Nox4 (green), Havcr1 (red) and; Lotus Tetragonolobus Lectin (LTL, white).

identified in the proximal tubule and accurately mapped these genes back to their original kidney compartments. Additionally, we provided a robust approach to integrate Visium and dRNA HybISS data, leading to accurate cell-type deconvolution of Visium spots. We revealed the dynamic changes of cell types and disease genes in 2D space and validated the spatiotemporal expression of disease targets. We used the gene imputation tools to investigate the spatial distribution and gene expression changes of the disease genes during kidney injury and repair. Finally, we report a spatial relationship between a C3+ monocyte subset and failed-repair proximal tubules. All of our raw and processed data, analysis pipeline, and computational tutorial are provided in publicly available data and code sharing repositories.

Kidney dissociation or nuclear isolation followed by scRNA-seq has become a standard approach for studying kidney cell-type diversity[1,32,49], cell state heterogeneity[2,17,50,51] the distinct responses of cells to disease and drug treatments[31,52]. However, this approach is limited by the loss of spatial information, complicating inference of cell–cell communication within the disease niche. Cell dissociation itself also introduces stress-response gene expression artifacts that complicate downstream data interpretation[53]. Furthermore, both single cell and single nucleus dissociation introduces bias in cell representation[54]. dRNA HybISS overcomes these limitations and detects transcripts in situ without requiring tissue dissociation or nuclear isolation. This direct characterization of gene expression on a kidney section allows for a less biased analysis of cell distribution and gene expression. Furthermore, it enables the validation of important disease signatures and the study of physical cell–cell interactions without relying on computational predictions. Our study provides a proof of principle for investigating cell diversity in both healthy and diseased kidneys, complementing other widely used single-cell technologies.

Sex dimorphism in acute kidney injury (AKI) has been observed in both animal models and clinical settings[55]. In mouse models of renal ischemia-reperfusion (IR) injury, females have been observed to be more resistant to kidney injury compared to males[56]. A study conducted on C57BL/6 mice revealed that male mice showed higher levels of IR-induced tubular injury and macrophage infiltration, which was associated with increased expression of inflammatory cytokines such as tumor necrosis factor-α, monocyte chemotactic protein-1, interferon-γ, and chemokine (C-C motif) ligand 17[57]. Another study identified that Sirt3 might mediate sexual dimorphism in AKI[58]. The discovery of sex-biased gene expression in each kidney cell type can be crucial for understanding the sex dimorphism observed in acute kidney injury (AKI) and other kidney diseases. A prior snRNA-seq study aimed at comparing gene expression differences between sexes revealed numerous genes that vary between male and female kidneys, particularly within the PTS3 segment[35]. In this spatial transcriptomics study, we corroborated the spatial distribution of genes exhibiting sex differences. Using gene imputation, we verified additional genes associated with sex differentiation, underscoring the significance of understanding the spatial and molecular dimensions of sex-specific gene expression. This is pivotal for advancing our comprehension of sex dimorphism in renal physiology and diseases, potentially guiding the development of sex-specific therapeutic strategies.

Our previous study used Visium to characterize IRI yielded some important findings, including the identification of an enhanced immune–epithelial interaction during the repair phase of AKI[8]. However, this dataset was limited by low resolution which hindered our ability to further elucidate the detailed intercellular communication maps. In addition to Visium, a higher-resolution technique called slide-seq has also been used to reveal immune–tubule interactions[10]. However, this technique has limited power for

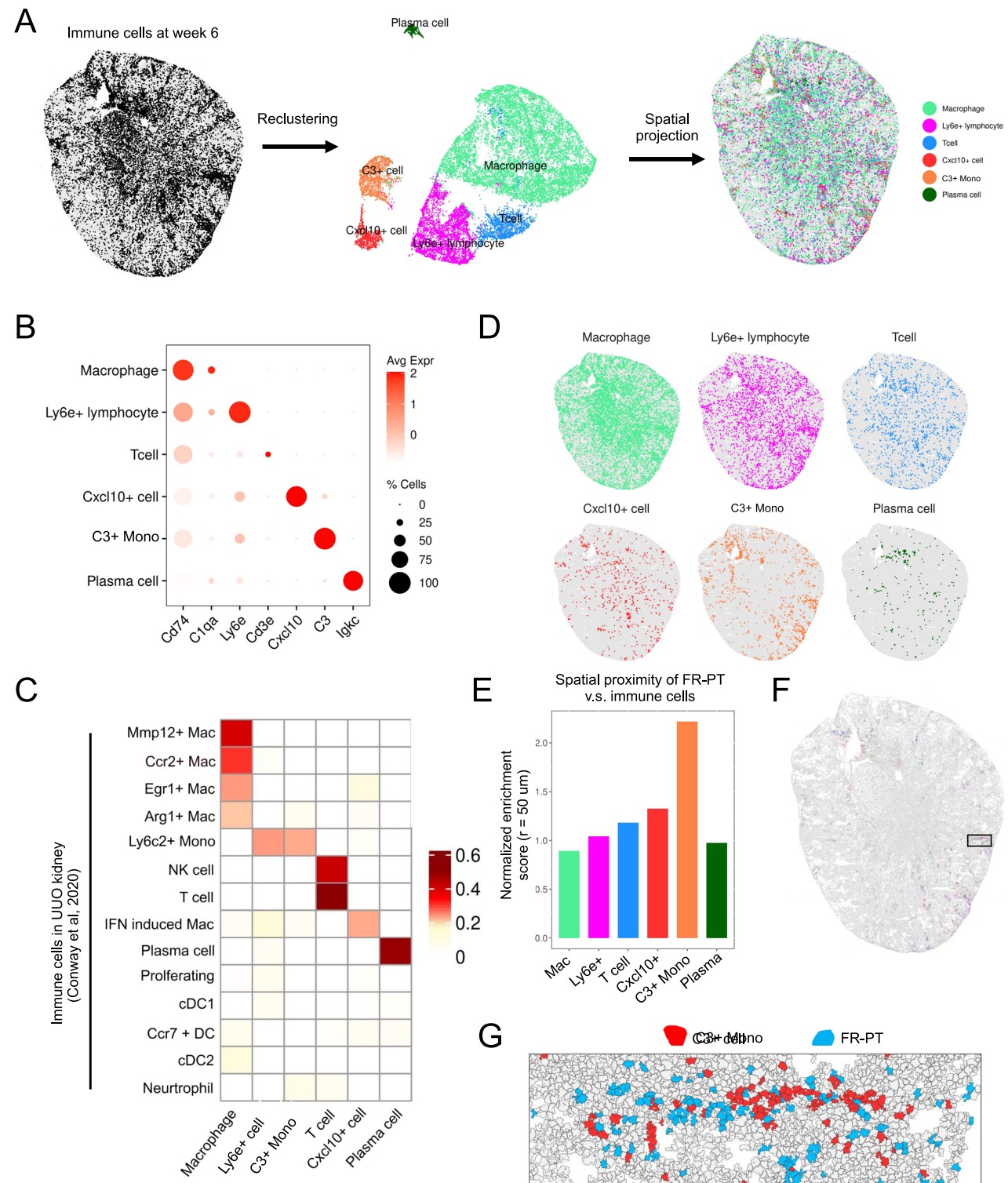

**Fig. 7 | Heterogeneity of immune cells in the recovery phase of AKI.**
**A** Classification of immune subsets at week 6. **B** Marker genes to define the immune cell identities. **C** Immune subtypes at week 6 of IRI were mapped to the immune cell types identified from the UUO kidney using Pearson correlation analysis. Mono monocytes, Mac macrophages, DC dendritic cells. **D** Spatial distribution of each immune cell type in the kidney. **E** Spatial proximity of failed-repair proximal tubular cells (FR-PT) and each immune subtype through cell-enrichment analysis. **F**, **G** Visualization of the C3+ immune cells and FR-PT in a selected kidney region.

detecting genes with low expression and rare cell types[10]. The high-resolution dRNA HybISS measurements of gene expression in single cells not only allowed us to detect low-expression genes like the podocyte markers and rare cell types like JGA cells, but also enabled us to chart the detailed spatial relationships between these rare cell types. For example, we identified that, of all the immune subtypes in week 6 of IRI, only the C3+ monocytes (a rare cell type) are in close proximity to the FR-PT cells (another rare cell type), suggesting active interaction between the two. This data precisely pinpoints the immune subtype that contributes to myeloid cell-mediated inflammation during initial kidney injury, a widely reported pathological event in the literature[59,60].

A limitation inherent in the dRNA HybISS technology applied in this study is its restriction to a panel comprising 200 genes, which allowed for the inclusion of 50 custom kidney genes in our probe design. Although this 200 gene panel incorporated 11 genes specific to lymphocytes, most of these were undetected in our analysis. As a result, we were unable to identify lymphocyte subpopulations. Similarly, neutrophils were also not identified in our dataset, likely due to an insufficient number of neutrophil-specific markers in our gene panel design. We would have needed a larger panel of immune subtype-specific genes to detect them. This limitation has now been addressed in more recent commercial versions of this technology, Xenium, which now allows for 480 custom gene probes. In addition, several tools from the computational side have also been developed to specifically address this issue[28–30,61]. A common practice of those tools is that they integrate the scRNA-seq data from matched samples and use the gene expression data from scRNA-seq to impute the missing genes in the ISS-based data. This can leverage ISS datasets to encompass the entire genome. We validated this approach to infer sexually dimorphic gene expression patterns and intercellular communication networks. The accuracy of those tools, however, still relies on the specificity of the genes being measured in the ground truth spatial dataset. As ISS technologies mature[14], this situation is expected to be alleviated.

In conclusion, we expanded the resolution of spatial transcriptomics in kidney research to a cellular level. The high-quality data generated by dRNA HybISS revealed fine kidney structures, cell marker expression patterns, and how the spatial expression patterns of the disease signature change during acute kidney injury. Our spatial tools and analytic approaches should be useful for future cell-resolution spatial transcriptomic analyses across tissues.

## Methods

### Animals
All in vivo experiments were performed on 8- to 10-week-old C57BL6/J male mice from The Jackson Laboratory. Experiments and housing guidelines were executed in accordance with the Animal Care and Use Committee at Washington University in St. Louis. Mice were maintained on ad libitum food and water in a 12-h light:dark cycle. The mouse housing room was maintained at humidity 30%–70% and temperature 20–26 °C (68–79 °F).

### Bilateral IRI surgery
Bilateral IRI (Bi-IRI) surgeries were performed as previously described[8,17,42]. Briefly, mice at 8–10 weeks old were anesthetized on isoflurane (1.8%–2%) with administration of buprenorphine for analgesia. Body temperature was kept between 36.5–37.5 °C on a heated pad and monitored by a rectal thermometer. Both kidneys were exposed by bilateral flank incisions made through the skin and the fascia. Both right and left kidneys were clamped with nontraumatic microaneurysm clamps (RS-5420, Roboz) for 23 min. The clamps were removed at time completion and kidneys were able to reperfuse at 37 °C (kidney color turned from dark red to pink). Kidneys were returned to the peritoneal cavity. The peritoneal layer of the skin was closed with absorbable suture and the flank incisions closed with wound clips. Mice were rehydrated by subcutaneous injections of warmed, sterile saline and recover in a 50 °C chamber before being reintroduced to animal facility. For tissue collection, mice were euthanized after Bi-IRI for sham control or 4 h, 12 h, 2 days, and 6 weeks post-surgery. Blood for end point kidney function analysis was taken from the inferior vena cava. BUN measurement at 24 h was done using the QuantiChrom Urea Assay kit for surgery quality control.

### Cartana library preparation and in situ sequencing
Fresh kidneys were incubated in 2-methylbutane (Millipore, Sigma) equilibrated in liquid nitrogen to maintain RNA integrity. Tissues were then stored at −80 °C and embedded in cryomolds using OCT (Tissue-Tek 4583) on dry ice. For frozen sectioning, OCT blocks were equilibrated to −18 °C, and 10 µm-thick sections were mounted onto the glass slides (1358W; Globe Scientific). Libraries were prepared using the high sensitivity (HS) library preparation kit provided by Cartana (part of 10X Genomics) per manufacturer's instruction. In brief, slides were fixed in freshly prepared formaldehyde diluted in 0.01% DEPC treated PBS, followed by permeabilization in 0.1 M HCl (DEPC treated). After dehydration and rehydration in series of alcohol gradients, slides were incubated in RM1-mix (provided by the library kit) overnight at 37 °C. Kidney sections were incubated in WB4 (library kit) at 37 °C for 30 min. Gene probes were ligated in ligases dissolved in RM2 (library kit) at 37 °C for 2 h. Transcripts were amplified in RM3-mix (library kit) at 30 °C overnight. Slides were stained with DAPI for quality control imaging and shipped to 10X Genomics for in situ sequencing.

### Visium library preparation and sequencing
After removing the renal capsule, the kidneys were bisected in a coronal manner to prepare for 10X Genomics Visium sample preparation. To preserve the high quality of RNA for processing, the fresh tissue was immersed in a bath of 2-methylbutane equilibrated with liquid nitrogen. The tissue samples were then stored at −80 °C prior to embedding them in optimal cutting temperature compound (OCT). The remaining kidney tissue was fixed overnight in 10% formalin at room temperature, before being transferred to 70% ethanol for storage at 4 °C. The tissue was subsequently embedded in paraffin by the Washington University Musculoskeletal Research Center Core. Frozen kidney samples were embedded in cryomolds using OCT on dry ice. The blocks were then stored at −80 °C. For preparation of Visium, the blocks were equilibrated to −18 °C, and 10 µm sections were mounted onto the active sequencing areas of the 10X Genomics Visium slides. The slides were stored in airtight containers at −80 °C until used for spatial library generation. Hemotoxylin and eosin staining were performed according to the 10X Genomics Visium protocol. Visium libraries were prepared according to 10X Genomics Visium manufacturer's instructions (PN-1000185, Lot No. 155614, Rev D). Sequencing was performed on a NovaSeq S4, targeting 125 million reads using dual indexing. Resulting FASTQ files were aligned to mm10 reference, manually aligned to respective hematoxylin and eosin stained images. The counts were normalized using 10X Genomics Space Ranger (spatial 3′ v1; spaceranger-1.2.1). The gene-by-count matrix was input to CellScopes.jl for downstream analysis.

### Cell segmentation for dRNA HybISS
We used Baysor[22] to assign the transcript imaging spots captured by dRNA HybISS into cells. We employed Baysor in prior mode, which requires nuclei segmentation as an initial input. We evaluated the performance of Watershed and Cellpose[21] for nuclei segmentation (based on DAPI staining). The Watershed algorithm was executed on the ImageJ platform (Fiji). The images were initially converted to 8-bit and processed with a Gaussian Blur filter. The threshold was adjusted using the default method, and then the images were further processed by the Watershed algorithm in the binary category. Cellpose was run on our Nvidia GPU server using default settings and a pre-trained nuclei model. The images after Cellpose segmentation were input into Baysor. Low-quality cells were filtered out using Baysor's cell segmentation statistics, such as number of transcripts per cell, elongation characteristics, cell area values, and average confidence scores of segmentation. Cells that did not meet the quality standards or were not located in the kidney were excluded from further analysis. We further filtered out the cells with less than five transcripts detected. The remaining segmented and filtered cells were used as input for downstream cell-type assignment analysis. To annotate the cells, we used the cluster labels from Baysor and defined the cluster

identity by inspecting the marker gene expression in each cluster. To validate the accuracy of our cell-type annotations, we employed two different methods. First, we compared our cell-type annotations to the cell types identified from Seurat clustering. We then calculated the Pearson's correlation and analyzed the proportion of cells overlapping within same cell types to evaluate the concordance between the cell types identified from Baysor and Seurat clustering. Second, we used the label transfer algorithm in Seurat V4 (https://satijalab.org/seurat/articles/integration_mapping.html) to map the cell types from snRNA-seq onto our dRNA HybISS dataset.

## CellScopes.jl for spatial data analysis

We created a Julia package, CellScopes.jl, to aid in downstream analysis after cell segmentation. Installation can be done via the Julia package manager. CellScopes offers simple usage, allowing users to directly call functions for data processing, normalization, and visualization. The current version supports data types from single-cell profiling techniques such as scRNA-seq and scATAC-seq, and spatial profiling techniques such as Visium, Slide-Seq V2, dRNA HybISS, Xenium, MERFISH and STARmap. We also customized CellScopes.jl for kidney spatial analysis, including a cell-centric approach to calculate cell–cell distance and a kidney coordinate system to depict cell location and gene expression changes along the kidney axis (See details methods below). To facilitate the transition from other languages to Julia, we have enhanced CellScopes's accessibility by providing functions that can directly read Scanpy AnnData and Seurat R objects, converting them to corresponding objects within CellScopes.jl (https://github.com/TheHumphreysLab/CellScopes.jl).

## Spatial proximity analysis

To reveal the physical cell–cell contact, we provided two approaches to calculate the distance of any given cell populations depending on their cell distribution patterns. (1) When the cells are confined to some specific regions (such as the glomerular cell types), we take a cell-centric approach to calculate the cell–cell distance. We take each cell from the cell population of interest, and compute the distance between this cell and the cells from other cell types. This process will be iteratively repeated until all cells from the cell type of interest are done. For example, this can be applied to measure which EC subtype is close to the podocyte within a given search area (radius). (2) When the cell distribution is very diffusive (such as the immune cells or fibroblasts), we use a cell-enrichment approach to compute the spatial proximity of pairs of cell types as reported by Lu et al.[48]. We calculate the probability of cell-type pairs in a neighborhood within a given searching area. We then compute the enrichment of cell-type pairs in spatial proximity after normalized to the control probability based on random pairing. These approaches were incorporated in CellScopes.jl.

## Kidney coordinates to study the gene and cell distribution along the kidney axis

In CellScopes.jl, we created a new coordinate system, namely kidney coordinate system, to precisely depict the position of every single cell in the kidney. In this system, the position of a cell was defined by the kidney depth, and the kidney angle. To transform the xy coordinate system to kidney coordinate system, we first defined the origin of the coordinate by finding the center point in the papilla. For each cell, we computed the kidney depth by calculating the distance of the cell to the kidney boundary, and divided by the distance of the kidney boundary to the origin of the coordinate. We can define the kidney angle of the cells by measuring the angle of the slope and the new x coordinate (in tangent value) (Supplementary Fig. 11B). This kidney coordinate system can help define the kidney compartment where the cell resides, how the cell-type and transcript distribution changes from outer cortex to papilla, and how the gene expression changes in different conditions.

## Benchmark the analyses from CellScopes, Seurat V5, Giotto and Squidpy

In order to compare the output from CellScopes, Seurat, Giotto, and Squidpy, we downloaded two publicly available healthy human kidney datasets from the 10X website – one from Xenium (https://www.10xgenomics.com/resources/datasets/human-kidney-preview-data-xenium-human-multi-tissue-and-cancer-panel-1-standard) and the other from Visium (https://www.10xgenomics.com/resources/datasets/human-kidney-11-mm-capture-area-ffpe-2-standard). We first used CellScopes to process the data for cell clustering and annotation, then applied the same cell labels to the clustering outcomes from Seurat, Giotto, and Squidpy. Since CellScopes is the only tool that can select the Field of View (FOV) based on its grid system, we chose the FOV to visualize the kidney structure from the region of interest, and applied the same FOV (determined by the xy coordinates) to the other tools. For a more direct comparison, we maintained a consistent color coding for cell annotation, and visualized same genes on the same kidney region across all tools. In cases where a tool lacked certain visualization functions, those particular plots were not shown in the figure.

## Integration of dRNA HybISS and Visium

For registration of Visium to dRNA HybISS data, we aligned matched kidney sections vertically and rotated them to overlap the two graphs with similar kidney structure. CellScopes was used to perform this alignment and coordinate transformation, using some important functions such as *rotate_axis*, *align_coordinates*, and *split_field*. We then cropped the area that achieved the best alignment outcome based on the kidney structure and the spatial expression pattern of selected cell-type markers. Using the aligned graph, we binned cells and transcripts from ISS by dRNA HybISS into the Visium spots based on proximity. For instance, we identified the closest spot to a cell or transcript as the Visium spot where it lies within. As a result, the cell-type composition of each Visium spot can be directly measured by the cell-type information obtained from the overlaid ISS data in the same area.

## Immunofluorescence staining

6 μm cryosections from frozen OCT blocks were cut and mounted onto slides. Kidney sections was fixed using 4% paraformaldehyde in 1× PBS for 10 min, washed with 1× PBS (3 times, 5 min each), and then blocked with blocking buffer (1% BSA, 0.1% Triton X-100 in 1× PBS) for one hour. Primary antibodies Nox4 (PA5-95083, ThermoFisher; 1:100), Havcr1 (AF1817, R&D Systems; 1:100), and lotus tetragonolobus lectin (B-1325-2, Vector Labs; 1:500) were incubated overnight at cold room (4 °C). Samples were washed in 1× PBS (3 times, each) and incubated with secondary antibodies Alexa Fluor® 488 anti-rabbit (711-545-152; Jackson ImmunoResearch; 1:200), Alexa Fluor® 568 anti-goat (A11057, Fisher Scientific, 1:200), and Alexa Fluor® 647 anti-streptavidin (016-600-084, Jackson ImmunoResearch, 1:1000) for one hour at room temperature in the dark. Sections were stained with 4′,6-diamidino-2-phenylindole (DAPI) and mounted with Prolong Gold (Life Technologies). Images were obtained by confocal microscopy (Nikon C2+ Eclipse; Nikon, Melville, NY).

## Visium cell-type deconvolution

We utilized our prior healthy kidney snRNA-seq dataset as a reference to determine the cell-type composition of each Visium spot using various publicly available tools, including Spotlight[38], RCTD[39], TACCO[40], and STdeconvolve[41]. We compared the results obtained from the deconvolution tools to the cell-type proportions directly measured by dRNA HybISS. We calculated Pearson correlation

coefficients to assess the accuracy of each tool in estimating the cell-type proportions.

## Genome-wide imputation of expression in dRNA HybISS

To assess the robustness of the spatial gene imputation tools in our ISS data, we benchmarked the performance of Tangram, SpaGE and gimVI using our matched timepoint snRNA-seq dataset[17]. We then used two approaches to estimate the accuracy of the imputation result. First, we selected a gene with a known expression pattern (such as the principal cell marker Aqp4), which was included in the probe design, and performed a leave-one-out analysis. In brief, we withheld expression value of Aqp4 from the dataset and used the imputation tools to predict the expression of Aqp4. We then inspected the spatial expression of Aqp4 by comparing the measured value and imputed value. Second, we chose a gene that has cell-type-specific expression pattern based on the snRNA-seq data and has not been included in our probe design. In this case, we used a new PTS3 marker Wdr4. Then we used the imputation tools to infer the expression of the Wdr4 and inspect its spatial expression in the tissue.

## Re-analysis of the public scRNA-seq datasets

We used Seurat to reanalyze publicly available scRNA-seq data from initial data processing to final cell clustering. To mitigate batch effects for datasets with multiple samples, such as those from Ransick et al.[35] and Conway et al.[47], we employed a data integration workflow as described in the Seurat tutorials (https://satijalab.org/seurat/articles/integration_introduction.html). Cell clusters were annotated based on known cell-type markers and correlation with our previous snRNA-seq dataset[17], and a scRNA-seq dataset published by other laboratory[33]. Cell-type mapping and annotation were performed using the Seurat label transfer tutorial provided by Satija Lab (https://satijalab.org/seurat/articles/integration_mapping).

## Subclustering analysis of the immune cell population at week 6 of IRI

We extracted the gene expression data for the immune cells from week 6 and re-clustered them using Seurat. We annotated the immune subtypes by mapping the clusters in our spatial data to the public kidney immune cell types idnetified from a scRNA-seq dataset[47].

## Statistics and reproducibility

Statistical analysis was conducted on all collected samples and data. No statistical method was used to predetermine sample size. No data were excluded from the analyses. The experiments were not randomized. Investigators were not blinded to allocation during experiments and outcome assessment.

## Reporting summary

Further information on research design is available in the Nature Portfolio Reporting Summary linked to this article.

## Data availability

All spatial transcriptomics data generated in this study have been deposited in the Gene Expression Omnibus (GEO) under accession number GSE227046. The dRNA HybISS (Cartana) data are available under accession number GSE227044. The Visium raw data can be accessed under accession number GSE227045. Source data are provided with this paper. Public single-cell RNA-seq data were collected from GEO with accession numbers: GSE180420, GSE139107, and GSE182256. The Xenium and Visium datasets for human kidney were downloaded from the 10X Genomics website (https://www.10xgenomics.com/resources/datasets?query=&page=1&configure%5BhitsPerPage%5D=50&configure%5BmaxValuesPerFacet%5D=1000). Source data are provided with this paper.

## Code availability

Original codes to reproduce the figures were deposited on GitHub at https://github.com/TheHumphreysLab/Spatial_analysis. A Julia package, CellScopes.jl, for analyzing various spatial transcriptomics data were available at https://github.com/TheHumphreysLab/CellScopes.jl. The codes are also available at zenodo (https://doi.org/10.5281/zenodo.10499323).

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

## Acknowledgements

This work was funded by seed network grant CZF2019-002430 from the Chan Zuckerberg Initiative and NIH grants DK103740 and 1U54DK137332 (all to B.D.H.).

## Author contributions

H.W. and B.D.H. conceived, coordinated, and designed the study. H.W. performed experiments and developed computational analysis with contributions from E.D., Q.X., J.G. and Y.Y. H.W., C.D., A.N., X.H., M.R. and B.D.H. analyzed data. H.W. and B.D.H. wrote the manuscript. H.W.,

B.D.H. and M.R. edited the manuscript. All authors read and approved the final manuscript.

## Competing interests

B.D.H. is a consultant for Janssen Research & Development, LLC, Pfizer, Roche and Chinook Therapeutics, holds equity in Chinook Therapeutics and grant funding from Pfizer, all unrelated to the current study. D.C., H.X., A.N. and M.R. were employees of 10X Genomics during this study. M.R. holds stock options in 10X Genomics. The remaining authors declare no competing interests.
