## [Peer Review File · Nature Communications]

REVIEWER COMMENTS

Reviewer #1 (Remarks to the Author):

This is an interesting and well performed study. The main advantage is the availability of the cartana instrument to the investigators.

The looks pretty nice. I think the team is a bit carried away with the interpretation.

How many cells can the authors actually identify? The team has 200 markers and the cell identification seems pretty good. There is a good amount of empty space on figure1a. If we count all dapi positive cells and compare to marker labelled cells, how does the data look?

Data QC in aggregate and in single sample should be shown.

Figure2. it is not clear whether this is just single sample comparison or do we have some sort of statistical testing of multiple samples and reproducibility and statistical estimates?

The markers used by the team not very typical for cell types?

Are these genes different when male and female samples compared in bulk mouse and human data?

Figure4. I am not sure that this is fair comparison. It is obvious that visium has lower resolution, but is genome wide data. Furthermore it is unclear whether the team was able to generate high quality visium data. Can they compare to 10x website visium FFPE data, after general pipeline, spotcleaning etc?

Yes this tool is really good for cell type identification, but can we detect gene expression changes between healthy and disease state or this tool is only to analyze cell fraction and location changes?

Could the team cluster the data with a known kidney scRNAseq data for example Balzer et al Nature Comm 2022?

The immune cell type identification is still not perfect, likely due to the low number of markers. How do these markers compare to other higher resolution kidney immune cell data by Doke et al Nature Immunology?

Overall this is an interesting technical paper

Reviewer #2 (Remarks to the Author):

In this paper, Wu et al apply spatially resolved transcriptomics via dRNA HyBISS to characterize kidney injury and repair in a murine model of kidney injury. While this paper presents an interesting application of a newer spatial transcriptomic technology to a spatially complex organ, the insights presented in the paper represents at best an incremental advance in its current form.

Biologically, there appears to be limited new insights regarding kidney injury in a murine model setting beyond that has already reported from the lab's previous work (<https://www.ncbi.nlm.nih.gov/pmc/articles/PMC8819997/>) and this other work <https://insight.jci.org/articles/view/147703>.

The authors do uncover sexually dimorphic genes in mice but the clinical relevance of these observations to humans remain to be discussed.

Computationally, the authors develop a new analysis pipeline CellScopes.jl. However, it is not clear whether CellScopes.jl will be applicable to other spatial transcriptomics technologies, particularly those based on in situ hybridization (such as MERFISH, SeqFISH, STARmap, or even 10X Xenium). Likewise, it is not clear how it compares to other processing and analysis tools.

Given that the authors generated both high and low resolution spatial transcriptomics data for the same tissues, they use these data to directly evaluate the accuracy of deconvolution of low resolution spatial transcriptomics. Likewise, given that the authors generated both targeted spatially resolved transcriptomics and full transcriptome single nucleus RNA-seq data for the same tissue, they were able to impute gene expression patterns not measured by the targeted approach. Based on these demonstrations, the data generated as part of this paper will undoubtedly be a useful

resource for both kidney biology and computational tool development and benchmarking. However, the paper in its current form is difficult to follow given its breadth and limited depth in both biology and computation.

Some specific comments are below:

1. The authors claim that "CellScopes.jl can also be used to analyze other data types (such as scRNA-seq, Xenium and Visium)" (Line 162) however this is not demonstrated in the paper.

2. In general, it is not clear how CellScopes.jl compares to existing spatial transcriptomics analysis toolkits like Seurat, Giotto, etc

3. Given other, larger consortia efforts like the Kidney Precision Medicine Project that are also performing spatial transcriptomics in kidney research, statements such as "we expanded the resolution of spatial transcriptomics in kidney research to a subcellular level" (line 379) seem grandiose. Likewise, it is not clear how the authors took advantage of the subcellular-level information that was available since everything was segmented into single cells anyway.

4. Generally, to suggest that this limited sampling of murine models represents an "atlas of kidney injury and repair" as suggested by the title seems to be an overstatement.

5. Based on the author's previous work, it was suggested that proximal tubules and T cells became more colocalized upon injury (<https://www.ncbi.nlm.nih.gov/pmc/articles/PMC8819997/>). These new results suggest that proximal tubules are recruiting C3+ monocytes. How do the authors reconcile these new and previous observations?

6. There is text in the methods that appear copied from elsewhere and no longer make sense in the context of the paper. For example (line 498) "The schematic below explains the coordinate
499 transformation. This kidney coordinate system can help define the kidney compartment where
500 the cell resides, how the cell type and transcript distribution changes from outer cortex to
501 papilla, and how the gene expression changes in different conditions. Here are some steps to
502 complete the transformation."

But there is no schematic below. Perhaps this was copied from the software documentation.

Reviewer #3 (Remarks to the Author):

In the manuscript by Wu et al. entitled “High resolution spatially resolved transcriptomic atlas of kidney injury and repair by RNA hybridization-based in situ sequencing,” the authors present subcellular spatial transcriptomics of a murine kidney model with cutting edge massively multiplexed in situ hybridization technology. The authors leverage this technology to spatially localize prior findings, such as sexually dimorphic gene expression. The biological insights hold minimal novelty, but the models serve well as a means to explore the proposed methodology and analysis pipeline. The authors craft a tool for data analysis in Julia, and use it to identified associations between tubular and immune cell types. The manuscript is well written, and potentially impactful provided the methodology and analysis techniques can be understood by the readership. Further, there exist important deficits in the cell type annotation and distribution because only a small portion of scRNAseq clusters were called on ISH and some of these cell types did not follow expected distributions. Not enough information is provided to determine if these are technical deficits related to the 10x platform or analytic deficits related to the new CellScopes tool developed. The critiques below are intended to help the authors enhance the utility of both the platform and analytic methodology for the scientific community.

Major Critiques -

1. Transparency of methods –A clear understanding of the raw data processing steps, normalization, cell segmentation, cell type annotation, and neighborhood clustering is important for this novel technology. For example, the Baysor framework is used to cluster the cells, but how? The reference cited presents Baysor as a cell segmentation tool, not a clustering tool.
2. Transparency of Code –End to end code is not provided. It was not possible to replicate the analysis from the raw data with the code provided. The real value of this manuscript is how the data was processed, segmented, annotated, and clustered. As an example, I could not find a “male” dataset provided or code on raw data processing which leads from the image to the “male” dataset.
3. QC metrics: Please provide QC metrics for sensitivity and specificity of cell type calling. What proportion of nuclei are discarded or imputed overall and for each cell type? When nuclei remained unsegmented, was there a rationale? Is imputation handled differently in PTs which are assumed to have homogeneous cross sectional cell types and CDs which have a mix of PCs and ICs? You may need orthogonal validation with immunofluorescence or another technology to provide additional support for the colocalization and cell type distribution identified.
4. Cell annotation –Cell annotation could be a key strength of this atlas manuscript and an important gift to the scientific community. More detail is required to realize such utility. From a high resolution scRNA-seq atlas, what cell types can be distinguished on ISH? Which clusters need to be merged?

Can you only interpret 13 cell types from dozens of scRNAseq clusters? What are the ISH cell type definitions based on the combination of markers and expression / segmentation thresholds? Which cells are defined by a single marker v. multiple markers? Which cells types are unable to be defined? For example, why are a limited set of immune cell types present (i.e. no neutrophils or macrophage/monocyte subtypes)?

5. Cell distribution should be clarified – Whether the kidney in Figure 2 s a healthy control is not explicitly stated. Assuming it is healthy, could the authors comment on the high amount of injured cortical PT and the near absence of healthy PTs (S3) in the outer medulla? In Figure 3, why might there be no injured PT when it is prevalent in Figure 2? In Figure 2A-4, the purple may be problematic as it refers to both TAL and CD-IC. The inner medulla seems to contain a high proportion of either ICs or TAL cells as compared to CD-PCs? Wouldn't IC's be interposed within the same tubular cross section with CD-PCs? The image suggests distinct CD-PC tubules and CD-IC tubules. If so, would the authors interpret this as a true biologic signal or an imputation error? If the purple color instead refers to TAL, could the descending thin limb or ascending thin limb of the inner medulla be misclassified as TAL? For Figure 2D, could you clarify the difference between the 2 gEC cell types? Also for Figure 2D, it appears CD-ICs are a prevalent cell in the cortex, multiple fold more prevalent than cortical PC's - is this the interpretation the authors intended and is it supported by the images of Figure 2A?

6. Context – (Lines 141 – 164) – Creating a software package for the community is commendable. However, the authors state that previous tools are insufficient to handle the high cellular density of the kidney and only plot gene expression as discrete points. Both R and python software ecosystems have tools that handle a high number of cells successfully, and the Satija lab recently released Seurat 5 which produces a visualization similar to figure 1C. From the tutorials and documentation, CellScopes appears robust and brings versatility by allowing spatial analysis in Julia; however, a comparison with Seurat 5 to highlight the strengths of CellScopes (beyond it being in Julia) would be valuable to the community, particularly since we would ask scientists to switch from Python and R to Julia.

7. Deconvolution - Deconvolution is only performed in the papilla with mainly CD-PCs present. Could you kindly evaluate the method in the cortex and medulla which have greater cell diversity? The stated rationale of cell type purity is actually a drawback, since this evaluation is biased to call fewer cell types.

Minor critiques:

1. Line 194 –Kindly clarify what is meant by “horizontal” in the kidney coordinate system. It seems to be an arbitrary point with relation to the image orientation, and not anatomically based.

2. Line 226 – I believe a panel might be missing from supplemental figure 5B

3. Line 303 – In Figure 6C, it is not clear what the authors mean by mapping the immune subtypes to the previous Conway study. Also, what is the reason behind the disparity between the 2 datasets? For example, are neutrophils not detected in the model or there are no neutrophil marker genes included in the panel? This can alter the interpretation of the results presented in this section.

4. Line 242 – The cell types of Figure 4B for dRNA HybISS do not agree with the ones presented in Figure 2. Could you clarify? In Figure 5, all samples had the same 13 cell types. Was the clustering performed in a combined dataset? If so, how was batch effect accounted for?

5. In the conclusion, line 330, they state “Finally, we report previously unrecognized heterogeneity among immune cells in kidney repair”. Could this be an overstatement? Particularly considering the lack of immune cells captured by the ISH technology?

6. Line 498 - Kidney coordinates to study the gene and cell distribution along the kidney axis methods: I could not find a schematic included, but one is alluded to in the text.

7. Line 537 - Re-analysis of the public scRNA-seq datasets methods: Please note the version of Seurat, because the most recent version incorporates algorithms such as harmony for batch correction. Seurat, by itself, does not account for batch correction as implied in the text. The application of an algorithm such as harmony is recommended.

8. Line 275 - AKI to CKD transition: It is interesting to find Immune and Fib deposition in the medulla (Supp Fig 7) and an apparent loss of TAL (Fig 5a), could the authors offer an interpretation for this injury pattern? Were cell-cell interactions present between the TAL and stromal cells? Can you resolve myofibroblasts?

Reviewer #1 (Remarks to the Author):

This is an interesting and well performed study. The main advantage is the availability of the cartana instrument to the investigators.

The looks pretty nice. I think the team is a bit carried away with the interpretation.

How many cells can the authors actually identify? The team has 200 markers and the cell identification seems pretty good. There is a good amount of empty space on figure1a. If we count all dapi positive cells and compare to marker labelled cells, how does the data look?

We thank the reviewer for this important comment. In order to keep the kidney structure intact, we followed the common practice for spatial data processing of relatively minimal cell filtering (we removed cells with less than 5 transcripts detected for example). We have now included a supplementary table to summarize the total number of nuclei versus cells identified in each sample to make this point clearer.

Sample	Total nuclei	Total cells
Sham_male	123769	78392
Hour4	119572	114658
Hour12	100781	93990
Day2	102392	101702
Week6	135106	119666
Sham_female	117814	88961

Data QC in aggregate and in single sample should be shown.

Thank you for this comment. We have added QC data as a supplementary figure in the revised manuscript (See also the figure below).

Figure 1. Number of transcripts per cell in individual or combined samples during the IRI time course.

Figure2. it is not clear whether this is just single sample comparison or do we have some sort of statistical testing of multiple samples and reproducibility and statistical estimates?

Thanks for this comment. While we only included a single sample for each timepoint, it is important to note that each of these samples comprises approximately 100,000 cells. Statistical estimates were performed at a single cell level not at the sample level.

The markers used by the team not very typical for cell types? Thank you for raising this point. We would argue that the markers we selected are in fact typical/canonical markers for kidney cell types in Figure2. For example, Podocalyxin (Podxl) for podocyte, Slc5a2 (SglT2) for proximal tubule, and Umod for Loop of Henle. The specificity of these markers has also been validated in our previous scRNA-seq datasets (These can be found on our data visualizer website, K.I.T.: <http://humphreyslab.com/SingleCell/>).

Are these genes different when male and female samples compared in bulk mouse and human data? We compared these cell type markers from Figure 2 between sexes using publicly available bulk mouse and human data. As shown by the new figure below, there are no significant differences.

Figure 2. Marker gene expression in bulk RNA-seq data. **A.** heatmap showing marker gene expression between sexes in a mouse kidney. **B.** expression of the same cell type specific markers were shown in a human kidney bulk RNA-seq data.

Figure 4. I am not sure that this is fair comparison. It is obvious that visium has lower resolution, but is genome wide data. Furthermore it is unclear whether the team was able to generate high quality visium data. Can they compare to 10x website visium FFPE data, after general pipeline, spotcleaning etc?

We fully agree that Visium has its own advantages including genome-deep capture, as the reviewer points out. Our previous work used Visium to elucidate insights regarding cell-cell communication during the IRI time course (Dixon et al, JASN 2022). However, Visium has limitations as well, including relatively low resolution - much lower than single cell resolution. The primary message from Figure 4 is that Cartana, with its cellular spatial resolution, can precisely classify kidney cell types, and that integrating high resolution coordinates and cell annotations with the whole transcriptome Visium datasets could present a strategy for leveraging the strengths of each modality. Regarding the quality concern raised by the reviewer, we have now conducted a comparative analysis between our Visium data and a kidney Visium dataset (FFPE) from the 10x Genomics website. While the number of detected genes per cell was lower in our study, this difference can primarily be attributed to our use of a lower read depth for sequencing (Figure 3A in this response letter). Notably, the gene expression profiles between the two samples exhibit a strong correlation (Pearson $r = 0.87$, Figure 3B in this response letter), indicating comparable data quality. Additionally, both datasets are capable of identifying similar cell types (Figure 3C in this response letter), and the gene expression within each of these cell types shows a high degree of correlation between the two datasets (Figure 3D in this response letter).

Figure 3. Comparison between Visium datasets from this study and 10x Genomics. **A.** Genes and reads detected per spot. **B.** Correlation of gene expression in Visium datasets between this study and 10x Genomics. **C.** Cell clustering and annotation. **D.** Pearson correlation of cell types identified in the Visium dataset from this study and 10x Genomics.

Yes this tool is really good for cell type identification, but can we detect gene expression changes between healthy and disease state or this tool is only to analyze cell fraction and location changes?

We thank reviewer for pointing out this important aspect. As demonstrated in Figures 4B and 4C, as well as in Supplementary Figure 15, our dataset is capable of detecting changes in gene expression between healthy and diseased states, as well as across different stages of acute kidney injury.

Could the team cluster the data with a known kidney scRNAseq data for example Balzer et al Nature Comm 2022?

In response to the reviewer's suggestion, we employed Seurat's label transfer approach to integrate our Cartana data with two publicly available scRNA-seq datasets (Balzer et al., Nature Communications 2022 and Kirita et al., PNAS 2020). As shown below, the cell types identified in our Cartana dataset align accurately with their corresponding cell types in both scRNA-seq datasets (Figure 4 in this response letter). This not only reinforces the high accuracy of Cartana data for identifying kidney cell types but also validates the precision of our cell type annotations. The new figure has been included in the updated manuscript (Supplementary Figure 8).

Figure 4. Integrating Cartana data with public scRNA-seq and snRNA-seq Data. **A.** Combined analysis of our spatial data and the scRNA-seq dataset from Balzer et al. (Nature Comms 2022). **B.** Combined analysis of our spatial data and the snRNA-seq dataset from Kirita et al. (PNAS 2020). Cells are color-coded based on annotations from the original publications, our Cartana annotations, or predicted cell labels using Seurat label transfer derived from the original annotations.

The immune cell type identification is still not perfect, likely due to the low number of markers. How do these markers compare to other higher resolution kidney immune cell data by Doke et al Nature Immunology?

We appreciate this comment and agree. We examined the expression of immune-related genes in our study by comparing them with the Doke et al. dataset (Figure 5 in this response letter). Our new data show a high degree of consistency with the gene expression patterns observed in our Cartana data. Specifically, Cd74 and C1qa are predominantly expressed in macrophages, Cd3e is uniquely expressed in T cells, Cxcl10 and C3 are found in monocytes, and Igkc is characteristic of B cells (Figure 5A in this response letter). A Pearson correlation analysis further validates that the cell types identified in our Cartana dataset closely map to the corresponding immune cell subtypes defined by Doke et al (Figure 5B in this response letter). Some discrepancies in identified cell types between the two datasets are likely attributable to differences in the disease types studied (CKD versus AKI). Ly6e is expressed across various immune subtypes in the Doke et al. dataset, yet correlation analysis confirms its specific association with T lymphocytes.

Figure 5. Comparison of the immune gene expression and immune cell types between Cartana and a public scRNA-seq dataset. **A.** Expression of immune genes from this study in the scRNA-seq dataset by Doke et al (Nature Immunology 2022). **B.** Correlation between immune subtypes identified in this study and those from Doke et al.

Overall this is an interesting technical paper
Thanks very much for this positive comment!

Reviewer #2 (Remarks to the Author):

In this paper, Wu et al apply spatially resolved transcriptomics via dRNA HybISS to characterize kidney injury and repair in a murine model of kidney injury. While this paper presents an interesting application of a newer spatial transcriptomic technology to a spatially complex organ, the insights presented in the paper represents at best an incremental advance in its current form.

Biologically, there appears to be limited new insights regarding kidney injury in a murine model setting beyond that has already reported from the lab's previous work

(<https://www.ncbi.nlm.nih.gov/pmc/articles/PMC8819997/>) and this other work <https://insight.jci.org/articles/view/147703>.

Both of the cited articles are based on low resolution Visium datasets. In this application of dRNA HybISS to study kidney disease at cellular resolution, we included genes known as markers for specific cell types, as well as ischemia-reperfusion injury (IRI) disease markers, based on our previous snRNA-seq dataset (Kirita et al., PNAS 2020). Our aim was to evaluate the ability of dRNA HybISS to accurately identify kidney cell types and to map the spatial distribution of both disease-related genes and cell types. Given the constraints on the number of genes that could be included in the study (200 genes in total), it was challenging to use a data-driven approach to uncover a wide array of new biological insights. We argue that, with this high-plex hybridization spatial technique, we advance existing knowledge by identifying that Vcam1 FR-PT interacts specifically with a particular subset of macrophages—C3 monocytes (Figure 6), not all macrophages as the two prior studies reported. We also validate the spatial distribution/expression of a number of new disease genes such as Plin2, Gsta1, and Apoe (Figure 5C).

With the insights present from our current study and the new computational tool we have developed, future studies will be enabled through inclusion of alternative gene panels to delve deeper into biological phenomena using this advanced technology.

The authors do uncover sexually dimorphic genes in mice but the clinical relevance of these observations to humans remain to be discussed.

We now discuss the clinical relevance of the sexually dimorphic genes in greater detail within the Discussion section of the manuscript.

Computationally, the authors develop a new analysis pipeline CellScopes.jl. However, it is not clear whether CellScopes.jl will be applicable to other spatial transcriptomics technologies, particularly those based on in situ hybridization (such as MERFISH, SeqFISH, STARmap, or even 10X Xenium). Likewise, it is not clear how it compares to other processing and analysis tools.

CellScopes.jl is definitely applicable to a wide range of spatial transcriptomics technologies. To better communicate this adaptability, we have now added new functionalities and

comprehensive tutorials that guide users through the process of analyzing data from MERFISH, SeqFISH, STARmap, and 10x Xenium technologies (as illustrated in the figure below). All updated code and tutorials are publicly accessible in our GitHub repository (<https://github.com/TheHumphreysLab/CellScopes.jl>).

Figure 6. Spatial and single cell data types that have been incorporated in CellScopes.jl

Given that the authors generated both high and low resolution spatial transcriptomics data for the same tissues, they use these data to directly evaluate the accuracy of deconvolution of low resolution spatial transcriptomics. Likewise, given that the authors generated both targeted spatially resolved transcriptomics and full transcriptome single nucleus RNA-seq data for the same tissue, they were able to impute gene expression patterns not measured by the targeted approach. Based on these demonstrations, the data generated as part of this paper will undoubtedly be a useful resource for both kidney biology and computational tool development and benchmarking. However, the paper in its current form is difficult to follow given its breadth and limited depth in both biology and computation.

We thank the reviewer for these positive comments. We believe that our revised manuscript has been significantly improved after providing a large number of new analyses and improved CellScopes documentation, functionality and tutorials (see below). With new high-resolution platforms that have become available in the last six months (Xenium, CosMx) our demonstration of segmentation and our CellScopes package should be highly enabling for investigators.

Some specific comments are below:

1. The authors claim that "CellScopes.jl can also be used to analyze other data types (such as scRNA-seq, Xenium and Visium)" (Line 162) however this is not demonstrated in the paper. We now provide new, detailed tutorials for using CellScopes to analyze various single cell and spatial datasets include scRNA-seq, scATAC-seq, HybISS by Cartana, Visium, Xenium, MERFISH, Slide-seq, seqFISH, STARmap. This has been demonstrated in our GitHub

repository as well (<https://github.com/TheHumphreysLab/CellScopes.jl>). For the reviewer's convenience, we have also provided direct links to these tutorials (see below).

a. dRNA HybISS by Cartana:

https://github.com/TheHumphreysLab/CellScopes.jl/tree/main/docs/cartana_tutorial

b. scRNA-seq:

https://github.com/TheHumphreysLab/CellScopes.jl/tree/main/docs/scRNA_tutorial

c. scATAC-seq:

https://github.com/TheHumphreysLab/CellScopes.jl/tree/main/docs/scATAC_tutorial

d. 10x Visium:

https://github.com/TheHumphreysLab/CellScopes.jl/tree/main/docs/visium_tutorial

e. 10x Xenium:

https://github.com/TheHumphreysLab/CellScopes.jl/tree/main/docs/xenium_tutorial

f. MERFISH:

https://github.com/TheHumphreysLab/CellScopes.jl/tree/main/docs/MERFISH_tutorial

g. Slide-seq:

https://github.com/TheHumphreysLab/CellScopes.jl/tree/main/docs/SlideSeq_tutorial

h. seqFISH:

https://github.com/TheHumphreysLab/CellScopes.jl/tree/main/docs/seqfish_tutorial

i. STARmap:

https://github.com/TheHumphreysLab/CellScopes.jl/tree/main/docs/starmap_tutorial

To make CellScopes more accessible to the bioinformatic community who has been using R and Python, we offer straightforward ways that convert Scanpy and Seurat objects into CellScopes objects. This will allow users to seamlessly explore all features designed in CellScopes.

j. Conversion of Scanpy AnnData to CellScopes Objects:

https://github.com/HaojiaWu/CellScopes.jl/tree/main/docs/scanpy_conversion

k. Conversion of Seurat Objects to CellScopes Objects:

https://github.com/HaojiaWu/CellScopes.jl/tree/main/docs/seurat_conversion

2. In general, it is not clear how CellScopes.jl compares to existing spatial transcriptomics analysis toolkits like Seurat, Giotto, etc

In response to this valid concern, we now provide new data to directly compare the performance of CellScopes (Julia), Seurat (R), Giotto (R and python), and Squidpy (python) in processing a public human kidney Xenium dataset downloaded from the 10X Genomics website (<https://www.10xgenomics.com/resources/datasets/human-kidney-preview-data-xenium-human-multi-tissue-and-cancer-panel-1-standard>). The reason we opted to use a Xenium dataset is that toolkits like Seurat, Giotto and Squidpy are limited to reading input data only from specific spatial transcriptomics techniques that have been integrated into their systems. For instance, all packages offer a “read_xenium” function to import data directly from the 10x Genomics Xenium analyzer but they lack built-in functionality for unincorporated data types, such as the HybISS technique by Cartana used in this study. Based on this comparison, we generated five supplemental figures (Figures S2-S6). For reviewer’s convenience, we have also included the

figures in this response letter (Figure 7 – 10 below). Below are the summaries of the comparisons.

I. Unique Features of CellScopes.jl

a. Data Exploration

CellScopes.jl offers enhanced data exploration features absent in other tools. It allows users to draw a grid on the tissue space, assigning a number to each grid cell (Figure 7C in this response letter). This facilitates easy selection and zooming into specific fields of view (FOV) to examine intricate kidney structures. In contrast, Seurat and Giotto require users to manually input precise coordinates or pre-defined field of view (such as in MERFISH data) to achieve the same task, which is often cumbersome.

b. Gene Imputation

CellScopes.jl seamlessly integrates with popular gene imputation tools such as SpaGE, GimVI, and Tangram (https://github.com/HaojiaWu/CellScopes.jl/tree/main/docs/gene_imputation). It offers built-in functions to directly impute gene expression in imaging-based spatial techniques and visualize the results. Such functionalities are missing in Seurat, Giotto or Squidpy (Figure 7 - 10 in this response letter).

II. Specialized Analysis for Kidney Data

CellScopes.jl incorporates unique features tailored for kidney spatial data analysis. We developed a novel coordinate system that represents cell locations based on kidney depth and angle. This system enables whole tissue scanning of the kidney to measure cell distribution changes from the outer cortex to the papilla (Figure 2D). It also facilitates plotting gene expression changes over time and across space in disease conditions (Figure 5C).

III. Cell-Cell Distance Measurement

Moreover, CellScopes.jl provides two approaches for calculating cell-cell distances: the cell-centric and the cell-enrichment approaches. The cell-centric approach is particularly useful for analyzing cells confined to specific regions, such as the glomerular cell types in the kidney (Figure 2C). On the other hand, the cell-enrichment approach is more suited for measuring distances where cell distribution is diffuse, like the distance between immune cells and other cell types (Figure 6E). For a detailed tutorial, please refer to our Github page (<https://github.com/TheHumphreysLab/CellScopes.jl>). None of these specialized functionalities are available in Seurat, Giotto, or Squidpy.

The new figures have been included in the revised manuscript. We also added a new supplementary table (Supplementary table S2) to summarize the key advantages of CellScopes in contrast to the existing tools.

Figure 7. A public human kidney Xenium dataset analyzed by CellScopes.jl. **A.** Cell clustering and cell type annotation. **B.** Spatial projection of the cell labels. **C.** A grid system to facilitate the selection of field of view (FOV). **D.** Zoom in the selected FOV to visualize the cell labels. **E.** Zoom in the selected FOV to visualize the gene expression. **F.** Zoom in the selected FOV to visualize the transcript distribution of multiple genes. **G.** Visualizing cells with cell boundary in polygon format. **H.** Visualizing genes with cell boundary in polygon format. **I.** Visualizing genes after gene imputation.

Figure 8. A public human kidney Xenium dataset analyzed by Seurat V5 in R. **A.** Cell clustering and cell type annotation. **B.** Spatial projection of the cell labels. **C.** Zoom in the selected FOV to visualize the cell labels. **D.** Zoom in the selected FOV to visualize the gene expression. **E.** Zoom in the selected FOV to visualize the transcript distribution of multiple genes. **F.** Visualizing cells with cell boundary in polygon format. **G.** Visualizing genes with cell boundary in polygon format.

Figure 9. A public human kidney Xenium dataset analyzed by Giotto in R and python. **A.** Cell clustering and cell type annotation. **B.** Spatial projection of the cell labels. **C.** Zoom in the selected FOV to visualize the cell labels. **D.** Zoom in the selected FOV to visualize the gene expression. **E.** Zoom in the selected FOV to visualize the transcript distribution of multiple genes. **F.** Visualizing cells with cell boundary in polygon format.

Figure 10. A public human kidney Xenium dataset analyzed by Squidpy in python. **A.** Cell clustering and cell type annotation. **B.** Spatial projection of the cell labels. **C.** Zoom in the selected FOV to visualize the cell labels. **D.** Zoom in the selected FOV to visualize the gene expression.

3. Given other, larger consortia efforts like the Kidney Precision Medicine Project that are also performing spatial transcriptomics in kidney research, statements such as "we expanded the resolution of spatial transcriptomics in kidney research to a subcellular level" (line 379) seem grandiose. Likewise, it is not clear how the authors took advantage of the subcellular-level information that was available since everything was segmented into single cells anyway.

We agree with these points. Although HybISS by Cartana can provide subcellular level resolution, the focus of this manuscript was not on the subcellular distribution of mRNA transcripts. We have toned down this comment: "In conclusion, we expanded the resolution of spatial transcriptomics in kidney research to a cellular level."

4. Generally, to suggest that this limited sampling of murine models represents an "atlas of kidney injury and repair" as suggested by the title seems to be an overstatement.

We appreciate the reviewer's observation regarding the use of the term "atlas" in the title. We understand that the term may imply a comprehensiveness that our study limited to certain murine models, may not fully capture. The intent was to convey that we have collected a substantial and detailed set of data on kidney injury and repair (more than 500 K cells from 5 different IRI timepoints, for example). To address this, we changed the original title to more accurately reflect the scope of our work: "High resolution spatially resolved transcriptomic profiling of kidney injury and repair by RNA hybridization-based in situ sequencing."

5. Based on the author's previous work, it was suggested that proximal tubules and T cells became more colocalized upon injury

(<https://www.ncbi.nlm.nih.gov/pmc/articles/PMC8819997/>). These new results suggest that proximal tubules are recruiting C3+ monocytes. How do the authors reconcile these new and previous observations?

We appreciate the reviewer's attention to this crucial detail. The data in the current manuscript align with the observations from our prior study based on Visium. In that study, we observed changes in interactions between injPT and immune cells (T cells and macrophages/monocytes) during the IRI time course based on the deconvolution results from SPOTlight. The interaction strength between injPT and T cells amplified from early timepoints, such as hour 12, while the interaction with macrophages/monocytes peaked at week 6. The immunofluorescent staining data verified minimal costaining of macrophages/monocytes with injPT at hour 12, with a significant colocalization by week 6 (as shown in Figure 4D of our published Visium paper). The current manuscript corroborates these findings, showing monocytes colocalizing with injPT at week 6. A key distinction we would like to emphasize here is that while the Visium paper's insights were primarily derived from computational predictions due to Visium's lower resolution in identifying immune subtypes, our Cartana data offers a much higher resolution which allows us to directly classify the cell types and their relation to one another. The data in this study not only supports our Visium findings but adds new insights into the biology. For example, we have found that it is not all macrophages, but a subset of monocytes (C3+) that interact with injPT at week 6. This data underscores the capability of imaging-based ST to unveil novel biological insights into kidney repair after IRI.

That said, while we did have some lymphocyte markers in our CARTANA probes (Cd3, Cd4 and Cd8), we did not detect any of these transcripts. We now clarify in the results that we did not detect lymphocyte transcripts, and that this likely reflects that we had too few lymphocyte-specific markers that were detectable.

6. There is text in the methods that appear copied from elsewhere and no longer make sense in the context of the paper. For example (line 498) "The schematic below explains the coordinate 499 transformation. This kidney coordinate system can help define the kidney compartment where

500 the cell resides, how the cell type and transcript distribution changes from outer cortex to 501 papilla, and how the gene expression changes in different conditions. Here are some steps to

502 complete the transformation."

But there is no schematic below. Perhaps this was copied from the software documentation.

We thank reviewer for this important note. We have revised the text to include the schematic in the figures.

Reviewer #3 (Remarks to the Author):

In the manuscript by Wu et al. entitled “High resolution spatially resolved transcriptomic atlas of kidney injury and repair by RNA hybridization-based in situ sequencing,” the authors present subcellular spatial transcriptomics of a murine kidney model with cutting edge massively multiplexed in situ hybridization technology. The authors leverage this technology to spatially localize prior findings, such as sexually dimorphic gene expression. The biological insights hold minimal novelty, but the models serve well as a means to explore the proposed methodology and analysis pipeline. The authors craft a tool for data analysis in Julia, and use it to identify associations between tubular and immune cell types. The manuscript is well written, and potentially impactful provided the methodology and analysis techniques can be understood by the readership. Further, there exist important deficits in the cell type annotation and distribution because only a small portion of scRNAseq clusters were called on ISH and some of these cell types did not follow expected distributions. Not enough information is provided to determine if these are technical deficits related to the 10x platform or analytic deficits related to the new CellScopes tool developed. The critiques below are intended to help the authors enhance the utility of both the platform and analytic methodology for the scientific community.

We thank reviewer for these constructive comments. Due to limitations on the total number of genes that could be included in the study, we selected genes serving as established markers for kidney cell types or diseases based on our previous snRNA-seq dataset (Kirita et al, PNAS 2020). This allows us to assess the resolution and validity of our new technique against existing knowledge in the field.

Although somewhat limited in number, we did incorporate some novel genes for validation purposes. For instance, Figure 5C reveals the spatial distribution of stage-specific disease markers such as *Plin2*, *Gsta*, and *ApoE*. In addition, we also present biological insight, for example, Figure 6 uncovers a unique interaction between C3+ monocytes and the failed-repair proximal tubules (PT), rather than other macrophage subtypes. Our prior work had established that FR-PTC are characterized by a pro-inflammatory and pro-fibrotic phenotype, and ligand receptor analyses suggested that this PT cell state would recruit immune cells. But this was an inference not direct evidence. This spatial dataset shows that the persistence of FR-PTC does indeed drive local inflammation.

With respect to the reviewer’s concern about a small number of cell clusters being detected in our dataset, we would emphasize that our Cartana data is capable of identifying nearly all major kidney cell classes, including those in the glomerulus (Podo and gEC), tubular epithelia (PT, TAL, DCT, PC, and IC), tubular interstitium (Fib and immune), urothelium (Uro), and endothelium (aEC). However, our Cartana data does not distinguish subpopulations as effectively as scRNA-seq. This limitation is, again, largely due to the constraint of including only 200 genes in a single run where we need to prioritize the selection of both cell-type specific and

disease markers. An exception to this limitation is our ability to identify six different immune subtypes. This is attributed to our initial experimental design, in which we incorporated two panels of immune cell markers. This allowed us to focus on the interactions between various immune subtypes and the failed-repair proximal tubules (FR-PT).

Major Critiques -

1. Transparency of methods – A clear understanding of the raw data processing steps, normalization, cell segmentation, cell type annotation, and neighborhood clustering is important for this novel technology. For example, the Baysor framework is used to cluster the cells, but how? The reference cited presents Baysor as a cell segmentation tool, not a clustering tool. We thank reviewer for highlighting this crucial aspect. We have now incorporated additional information into the manuscript to provide a more comprehensive description of our data processing procedures from normalization to final cell type annotation. Additionally, we have clarified that the Baysor framework not only performs cell segmentation but also cell clustering. These additional methodologic details are now included in the Results and Methods sections of the revised manuscript.

2. Transparency of Code –End to end code is not provided. It was not possible to replicate the analysis from the raw data with the code provided. The real value of this manuscript is how the data was processed, segmented, annotated, and clustered. As an example, I could not find a “male” dataset provided or code on raw data processing which leads from the image to the “male” dataset.

Thank you for this comment. In the revised version, we have substantially updated the CellScopes package to enhance its accessibility and reproducibility. Here are the primary updates made to CellScopes:

1. We have added more examples demonstrating how CellScopes can be utilized to analyze various data types. All data chosen for these examples are publicly accessible and can be directly downloaded.
2. We have created distinct, in-depth tutorials for each data type. These tutorials guide users through the specific analyses related to each data type.

a. dRNA HybISS by Cartana:

https://github.com/TheHumphreysLab/CellScopes.jl/tree/main/docs/cartana_tutorial

b. scRNA-seq:

https://github.com/TheHumphreysLab/CellScopes.jl/tree/main/docs/scRNA_tutorial

c. scATAC-seq:

https://github.com/TheHumphreysLab/CellScopes.jl/tree/main/docs/scATAC_tutorial

d. 10x Visium:

https://github.com/TheHumphreysLab/CellScopes.jl/tree/main/docs/visium_tutorial

e. 10x Xenium:

https://github.com/TheHumphreysLab/CellScopes.jl/tree/main/docs/xenium_tutorial

f. MERFISH:

https://github.com/TheHumphreysLab/CellScopes.jl/tree/main/docs/MERFISH_tutorial

g. Slide-seq:

https://github.com/TheHumphreysLab/CellScopes.jl/tree/main/docs/SlideSeq_tutorial

h. seqFISH:

https://github.com/TheHumphreysLab/CellScopes.jl/tree/main/docs/seqfish_tutorial

i. STARmap:

https://github.com/TheHumphreysLab/CellScopes.jl/tree/main/docs/starmap_tutorial

3. As requested by the reviewer, to reproduce the figures presented in our manuscript, we have established a separate GitHub repository containing all the necessary code to generate these figures.

https://github.com/TheHumphreysLab/Spatial_analysis

The data produced in this study, including the "male" dataset, has been uploaded to GEO under the accession number GSE227046. The data will be made publicly available upon the acceptance of the manuscript. For the purposes of review, reviewers can access and download the data using the provided token: qpyvmimmzhmdlgr.

3. QC metrics: Please provide QC metrics for sensitivity and specificity of cell type calling. What proportion of nuclei are discarded or imputed overall and for each cell type? When nuclei remained unsegmented, was there a rationale?

Here is the metric of nuclei and filtered cells:

Sample	Total nuclei	Total cells
Sham_male	123769	78392
Hour4	119572	114658
Hour12	100781	93990
Day2	102392	101702
Week6	135106	119666
Sham_female	117814	88961

In line with the practices of many tools used in the analysis of imaging-based spatial transcriptomic data (such as Seurat, which only removed cells with zero transcripts), we performed only minimal filtering on cells (e.g. removing cells with less than 5 transcripts). This decision was made because imposing a higher threshold on cell filtering would eliminate too many cells from the dataset, thereby damaging the spatial structure of the tissue by creating excessive hollow areas. Despite this minimal cell filtering, the accuracy of cell clustering remains unaffected. This is evidenced by the expression of specific markers and the expected spatial distribution of cell types shown in Figure 2.

Is imputation handled differently in PTs which are assumed to have homogeneous cross sectional cell types and CDs which have a mix of PCs and ICs? You may need orthogonal validation with immunofluorescence or another technology to provide additional support for the colocalization and cell type distribution identified.

We have not observed significant differences in the handling of gene imputation between PTs and CDs. To illustrate the robustness of gene imputation in both PT and CD subtypes, we have

included a figure in the supplement where we showed the imputed gene expression in PT and CDs defined by our Cartana data and the scRNA-seq data. The results showed that gene imputation is overall accurate in PT and CDs (Figure 11 in this response letter). Additionally, in response to the reviewer's suggestion, we have validated the spatial expression change of Nox4 from gene imputation across IRI timecourse using immunofluorescent staining (Figure 12 in this response letter, and new Figure 6 in the main manuscript). Specifically we show downregulation of Nox4 protein in the middle timepoints, as predicted. This orthogonal approach lends further support to accuracy of the gene imputation.

Figure 11. Gene imputation for the PT and CD celltypes. **A.** Imputed gene expression of the PT, PC, and IC in our Cartana dataset. **B.** Measured gene expression in the snRNA-seq dataset (Kirita et al, PNAS 2020). **C.** Spatial expression of the imputed gene in our Cartana data.

Figure 12 (New Figure 6 in main manuscript). Immunofluorescence staining for gene imputation validation. Timecourse samples were stained with Nox4 (green, one of the imputed genes), Havcr1 (red), LTL (white) and DAPI (blue).

4. Cell annotation –Cell annotation could be a key strength of this atlas manuscript and an important gift to the scientific community. More detail is required to realize such utility. We have now included many more details in the Methods regarding how we annotate the cell types.

From a high resolution scRNA-seq atlas, what cell types can be distinguished on ISH? We performed label transfer in Seurat to transfer the label from single cell RNA-seq to Cartana data (Figure 4 in this response letter). The majority of the cell types can be identified by our Cartana data except the cell type subtypes such as different TAL populations.

Which clusters need to be merged? We merged clusters when they shared similar molecular signature and no distinct markers can be identified among those clusters. We have provided a detailed script how we performed this process on our new GitHub page for figure reproduction (https://github.com/TheHumphreysLab/Spatial_analysis).

Can you only interpret 13 cell types from dozens of scRNAseq clusters? Due to the limited gene number in the design, we can only interpret 13 cell types. However, these 13 cell types represent the full range of kidney cell lineages, including tubular epithelia, glomerulus, tubulointerstitium, endothelium, and urothelium.

What are the ISH cell type definitions based on the combination of markers and expression / segmentation thresholds? Which cells are defined by a single marker v. multiple markers? We used the expression of cell type specific markers to define the cell type on ISH. The markers used to define each kidney cell types were shown in Figure 2B.

Which cells types are unable to be defined? The cell types that were not defined include parietal epithelial cells (PEC), thin limb of loop of Henle (ATL-DTL) and the subpopulations within a cell type (e.g. mTAL, cTAL1 and cTAL2 were not distinguishable in the TAL cell cluster).

For example, why are a limited set of immune cell types present (i.e. no neutrophils or macrophage/monocyte subtypes)? We were unable to detect neutrophils in our data because the gene markers specific to them were not incorporated into our gene panel design. However, we did identify various macrophage/monocyte subtypes, including macrophage, Ly6e+ monocytes, and C3+ monocytes (Figure 6C).

5. Cell distribution should be clarified – Whether the kidney in Figure 2 s a healthy control is not explicitly stated. Assuming it is healthy, could the authors comment on the high amount of injured cortical PT and the near absence of healthy PTs (S3) in the outer medulla? In Figure 3, why might there be no injured PT when it is prevalent in Figure 2?

The kidney shown in Figure 2 is not healthy but rather taken from the Day 2 after injury time point. We selected a Day 2 kidney for inclusion in the figure for several reasons, all aimed at highlighting the capabilities of the HybISS technique by Cartana:

1. Comprehensive Cell Types: The Day 2 IRI kidney contains a broad spectrum of cell types, including those affected by disease, such as injured proximal tubules (PT). Given that our study spans five different time points, it is important to showcase a specimen that includes diseased cell types as part of a time-course analysis.
2. Representative Anatomy: The Day 2 IRI kidney possesses all the typical structural elements of a kidney, including the renal artery and urothelium. This makes it a perfect sample to

demonstrate the high resolution that the HybISS technique can achieve in revealing kidney structures.

In Figure 2A-4, the purple may be problematic as it refers to both TAL and CD-IC. The inner medulla seems to contain a high proportion of either ICs or TAL cells as compared to CD-PCs? Wouldn't IC's be interposed within the same tubular cross section with CD-PCs? The image suggests distinct CD-PC tubules and CD-IC tubules. If so, would the authors interpret this as a true biologic signal or an imputation error? If the purple color instead refers to TAL, could the descending thin limb or ascending thin limb of the inner medulla be misclassified as TAL? We have now changed the color for CD-IC to make it distinguishable to TAL (please see the updated Figure 2). The inner medulla contains a high proportion of cells from TAL, whereas CD-IC is interposed within the same tubule with CD-PC (Figure 13A in this response letter). Our Cartana data cannot classify the Loop of Henle subtypes since those subtype specific markers were not included into our gene list. Label transfer results indicate that the TAL contains minimal contamination of the thin limb cells (6%, Figure 13B in this response letter).

Figure 13. Spatial relationship of TAL/PC/IC cell types and cross-validation of the TAL annotation. **A.** Spatial distribution of TAL, CD-PC and CD-IC in tissue. **B.** Seurat cell type label transfer for mapping the cell types from snRNA-seq to the TAL cluster annotated in our Cartana data.

For Figure 2D, could you clarify the difference between the 2 gEC cell types? We apologize for the oversight. We mistakenly use “gEC” to label the JGA cluster. This has been corrected in the revised manuscript.

Also for Figure 2D, it appears CD-ICs are a prevalent cell in the cortex, multiple fold more prevalent than cortical PC's - is this the interpretation the authors intended and is it supported by the images of Figure 2A? In our current dataset, we are unable to differentiate between cortical PC and papillary PC. As a result, it may appear that CD-IC is more prevalent in the cortex compared to PC. This is because our annotated CD-PC includes PC cells from both the cortex and papilla. Figure 2A further supports this observation, where the PC cells, represented in cyan, are found in the cortex, medulla, and papilla.

6. Context – (Lines 141 – 164) – Creating a software package for the community is commendable. However, the authors state that previous tools are insufficient to handle the high cellular density of the kidney and only plot gene expression as discrete points. Both R and python software ecosystems have tools that handle a high number of cells successfully, and the Satija lab recently released Seurat 5 which produces a visualization similar to figure 1C. From the tutorials and documentation, CellScopes appears robust and brings versatility by allowing spatial analysis in Julia; however, a comparison with Seurat 5 to highlight the strengths of CellScopes (beyond it being in Julia) would be valuable to the community, particularly since we would ask scientists to switch from Python and R to Julia.

Available spatial tools were designed only to analyze those spatial datasets that have been incorporated in their packages. For example, Seurat v5 offers a “read_xenium” function to directly read the output from 10X Analyzer. However, they cannot process spatial modules not integrated within them, such as HyISS by Cartana in this study. This limitation is what drove us to develop CellScopes for the community, which is designed to analyze a wide range of spatial modules with similar data structures. As requested, to compare the data generated by CellScopes and other tools, we downloaded a human kidney dataset generated by Xenium and another human kidney dataset generated by Visium since all the popular spatial tools can analyze these two spatial data types.

Benchmarking analysis of different tools on the Xenium dataset generated 5 supplementary figures (Supplementary Figure S2 – 6 in the main manuscript, or Figure 7 – 10 in this response letter). While the reviewer rightly points out that Seurat V5 can replicate most of the figures produced by CellScopes, especially when compared to Squidpy and Giotto (Figure 7 – 10 in this response letter), we want to highlight that CellScopes still possesses distinct features that set it apart and perhaps above other tools. For example, CellScopes.jl allows users to draw a grid on the tissue space, assigning a number to each grid cell (Figure 7C in this response letter). This facilitates easy selection and zooming into specific fields of view to examine intricate kidney structures. In contrast, other tools require users to manually input precise coordinates to achieve the same task, which is often cumbersome. In addition, CellScopes.jl offers built-in functions to directly impute gene expression in imaging-based spatial techniques with popular gene imputation tools such as SpaGE, GimVI, and Tangram, and directly visualize the imputation results (Figure 7I in this response letter). Moreover, CellScopes.jl provides a novel

coordinate system that enables whole tissue scanning of the kidney to measure cell distribution changes from the outer cortex to the papilla (Figure 2D). It also facilitates plotting gene expression changes over time and across space in disease conditions (Figure 5C). Finally, CellScopes.jl provides two approaches for calculating cell-cell distances: the cell-centric and the cell-enrichment approaches. The cell-centric approach is particularly useful for analyzing cells confined to specific regions, such as the glomerular cell types in the kidney (Figure 2C). On the other hand, the cell-enrichment approach is more suited for measuring distances where cell distribution is diffuse, like the distance between immune cells and other cell types (Figure 6E).

For Visium data analysis, CellScopes.jl is able to incorporate any high resolution images as the background layer that allows colocalizing the cell type annotations or gene expression on top of the histological feature (Figure 14 in this response letter). This can significantly facilitate the interpretation of the cell type annotation and gene expression pattern by showing the delicate kidney structure (such as the glomerulus) in histological staining images or overlaying the protein expression in the immunofluorescent staining images with the mRNA detection (not shown). Other tools only allow importing the images from the 10X SpaceRanger output, which is not high resolution enough to visualize the kidney structure (Figure 14B in this response letter).

We have provided a supplementary table (Supplementary table S2) in this revised manuscript to highlight the key advantages of CellScopes in contrast to the existing tools.

We believe that all these functionalities will better serve the whole scientific community to interpret their spatial transcriptomics data.

Figure 14. A public human kidney Visium dataset analyzed by CellScopes, Seurat, Giotto and Squidpy. **A.** Cell clustering and cell type annotation by CellScopes. **B.** Visualization of PODXL expression in the same FOV using CellScopes, Seurat, Giotto, and Squidpy.

7. Deconvolution - Deconvolution is only performed in the papilla with mainly CD-PCs present. Could you kindly evaluate the method in the cortex and medulla which have greater cell diversity? The stated rationale of cell type purity is actually a drawback, since this evaluation is biased to call fewer cell types.

We appreciate the reviewer's concerns. Firstly, we actually performed deconvolution across the entire kidney section, which includes the cortex, medulla, and papilla, as shown in Supplementary Figure S14A-E. We have updated the text to clarify this. We chose papilla to

highlight given its simpler cell composition, consisting primarily of CD-PCs. This simplicity allowed us to easily detect inaccuracies in tools like Spotlight and RCTD, which predicted a broader array of cell types than expected. On the other hand, TACCO and STdeconvolve accurately detected that CD-PCs is the dominant cell type in papilla and a high correlation with the cell type composition decomposed by Cartana data (Supplementary Figure S14F). In response to the reviewer's query, we present the results for all spots in a heatmap format (Figure 15 of this response letter), which shows the correlation scores between each deconvolution tool and direct measurements. Our findings remain consistent: both TACCO and STdeconvolve outperform Spotlight and RCTD.

Figure 15. Correlation of the cell type composition in each Visium spot between direct measurement by Cartana and cell type deconvolution with Spotlight, RCTD, TACCO and STdeconvolve.

Minor critiques:

1. Line 194 –Kindly clarify what is meant by “horizontal” in the kidney coordinate system. It seems to be an arbitrary point with relation to the image orientation, and not anatomically based.

Thank you for bringing this to our attention. We have revised the text: "In this system, the position of a cell is defined by the kidney depth (measuring the distance of each cell type to the kidney capsule) and the kidney angle (measuring the slope of each cell with respect to the positive x-axis) (Supplementary Fig. 11B)."

2. Line 226 – I believe a panel might be missing from supplemental figure 5B
We used the missing panel to indicate that this gene was not included in the original gene panel design. We now put a “Gene not included in the panel” in the panel to avoid confusion.

3. Line 303 – In Figure 6C, it is not clear what the authors mean by mapping the immune subtypes to the previous Conway study. Also, what is the reason behind the disparity between the 2 datasets? For example, are neutrophils not detected in the model or there are no neutrophil marker genes included in the panel? This can alter the interpretation of the results presented in this section.

We cross-referenced the immune subtypes with the previous Conway study to ensure accurate cell type annotation, which accurately reflects the resident immune cells in the kidney. The absence of neutrophils in our dataset is likely because neutrophil marker genes like Ly6g and Cd11b were not included in our panel list. We have added more details to clarify this in the revised manuscript.

4. Line 242 – The cell types of Figure 4B for dRNA HybISS do not agree with the ones presented in Figure 2. Could you clarify? In Figure 5, all samples had the same 13 cell types. Was the clustering performed in a combined dataset? If so, how was batch effect accounted for?

In both Figure 4B and Figure 2, cell type annotation was performed based on the clustering result from a single sample. For Figure 5, we annotated cell types from a combined dataset after correcting for batch effects using the Seurat integration algorithm. This leads to slight differences in the cell type labels. It is noteworthy that distinctions arise primarily in the PT: while single-sample clustering can discern different PT subtypes, this difference was not captured when clustering the combined dataset. This is likely because the PT is the primary cell type affected in IRI, and the transcriptional difference between injured and healthy PT is more pronounced than differences among PT subtypes. We have expanded our Methods section to provide a clearer explanation of our clustering methodology.

5. In the conclusion, line 330, they state “Finally, we report previously unrecognized heterogeneity among immune cells in kidney repair”. Could this be an overstatement? Particularly considering the lack of immune cells captured by the ISH technology?
We thank the reviewer for bringing up this point. We have deleted this statement.

6. Line 498 - Kidney coordinates to study the gene and cell distribution along the kidney axis methods: I could not find a schematic included, but one is alluded to in the text.

The schematic is presented in supplementary Figure S11B. We apologize for any oversight and have clarified its location in the text of the revised manuscript.

7. Line 537 - Re-analysis of the public scRNA-seq datasets methods: Please note the version of Seurat, because the most recent version incorporates algorithms such as harmony for batch correction. Seurat, by itself, does not account for batch correction as implied in the text. The application of an algorithm such as harmony is recommended.

We appreciate the reviewer's comment. We have now specified in the methods section that we utilized Seurat V4. Since its third version, Seurat has incorporated batch correction (PMID: 31178118). We adhered to the tutorial provided by Satija's lab (https://satijalab.org/seurat/articles/integration_introduction.html) for this batch effect correction. As shown in Supplementary Figure S12B, the batch effect correction with Seurat is quite robust.

8. Line 275 - AKI to CKD transition: It is interesting to find Immune and Fib deposition in the medulla (Supp Fig 7) and an apparent loss of TAL (Fig 5a), could the authors offer an interpretation for this injury pattern? Were cell-cell interactions present between the TAL and stromal cells? Can you resolve myofibroblasts?

We fully agree that it is an interesting injury pattern to see a deposition of fibroblast/immune cells and a loss of TAL population in the medullary region. A possible interpretation is that medulla may be a primary site where kidney fibrosis occurs during AKI – CKD transition. In fact, the deposition of fibroblast in the medulla was also observed in our previous study (See the Figure 4D from Kirita et al, PNAS 2020 below). We were not able to study the cell-cell communication between TAL and fibroblast because the ligand-receptor pairs were not included in the design. Based on cell proximity analysis, we did not observe a close proximity between TAL and stromal cells (data not shown). We also checked the expression of injured markers in TAL and they were not upregulated compared to Sham. Due to the limitation on the gene number in the current Cartana design, we were not able to resolve the myofibroblast from the fibroblast cluster. We believe that all these interesting questions can be answered by designing a new gene panel that includes fibroblast/myofibroblast markers, TAL injury markers (based on snRNA-seq profiling), and ligand-receptor genes associated with TAL and fibroblast. These are all promising avenues of research that other researchers might pursue based on our current data.

D

REVIEWER COMMENTS

Reviewer #1 (Remarks to the Author):

The team did an excellent job in revising the manuscript. I have no further comments

Reviewer #2 (Remarks to the Author):

We commend the authors for their efforts in this thorough revision. We believe the datasets, computational tool, and benchmarks performed in this manuscript will be of interest and utility to the field and is suitable for publication.

Reviewer #3 (Remarks to the Author):

In the manuscript "High resolution spatially resolved transcriptomic profiling of kidney injury and repair by RNA hybridization-based in situ sequencing", the investigators resubmit a study using the Cartana platform to define cell type distribution in the murine and human kidney. Overall, the authors were responsive to several critiques, but there remain important clarifications needed. It is clear the authors care about providing high quality data and I will highlight the unaddressed critiques and remaining blindspots below to help them achieve this goal.

Main critique:

The authors employ a 200 gene panel, with most cell type definitions based on a single marker, yet only 13 cell types are identified, which is small number of cells identified. Some of these 13 cell types have unexpected distributions which remain unjustified. Depending on how cell types are counted, the authors appear to map more than 13 cell types, in some cases distinguishing PTS1, PTS2, and PTS3, as well as 6 immune subtypes (maybe you have 20 cell types?). After review of the raw datasets, there are even more markers which correspond to other cell types and it is unclear if the authors chose not to analyze these markers or if there were technical limitations of Cartana. One of 2 courses of action is suggested:

1. Re-run a subset of the samples with Cartana, optimizing probes for the problematic cell types to improve the confidence of and expand the number of cell types defined.
2. Present both the negative and positive data and address problematic cell types more clearly in the results. In the abstract, please clearly denote the number of cell types resolved on the Cartana platform and the high profile cells that could not be resolved. Add a section on limitations to the results detailing why important cell types failed.

Problematic cell types:

1. Lymphocytes – The approach to lymphocytes is not clear in this study. Could the investigators add clarity to the following statement on line 456: “although we included 3 lymphocyte genes in our probe panel, we could not detect any lymphocytes in our dataset”? How were you able to map T cells from the scRNAseq data set in Figure 7D? In the raw data on GEO, it appears that the following 11 lymphocyte genes are in the panel: Cd3e, cd3d, cd3g, cd8a, cd8b1, cd19, cd4, cd79a, cd79b, foxp3, ly6e. This is not counting JCHAIN or Igkc for plasma cells or NKG7 for NK cells or quite a few others (interleukins, NCAM1, etc). Thus, there are more than 3 lymphocyte genes. Did these probe sets fail or were they crowded out by more highly expressed genes? Basically, please include the data on your experience with all these immune cell markers in a limitations results section so readers of Nature Communications know how these classic markers of immune cells perform on the Cartana platform.
2. Vascular Smooth muscle cells – TAGLN appears to be included in your panel. Why is this excluded from the study? The raw data suggests it confidently mapped to certain cells in a separate cluster to that of the fibroblasts.
3. Proximal tubule – This study maps a different number of cell types in figure 2 (N=13), figure 3 (N=14), Figure 4B (N=15), Figure 4D (N=14), and Figure 5 (N=13). Sometimes, the PT-S1, S2, and S3 are distinguished, and sometimes the injured PT is removed. Then in Figure 7, the FR-PT is introduced which is not included in these prior figures. To help the reader, is it possible to provide 2 common sets of cell types, one based on unbiased Cartana clusters from merged/normalized samples and a second based on a common set of scRNA-seq label transfer clusters throughout for all specimens? If not, please justify why a different number of clusters is required. What genes define the FR-PT cell type? How did you include the slc5a12 probe set?
4. Intercalated cells – The intercalated cell distribution in this study is unexpected. In response Fig 13A, I see the interposed ICs with the PCs. However, the distribution of intercalated cells in the new Supp Fig 2-5 is abnormal - why are there multiple tubules composed almost entirely of intercalated cells? These are seen in the Upper third of the image in Supp Fig 2D/G, 3C/E, 4C/E and lower third in Supp Fig 5C (this one is upside down). Have these tubules been miscategorized? There are few if any principle cells colocalizing in these yellow tubules. Figure 2D still seems incorrect when comparing between rows. The proportion of ICs, JGA, and other lowly represented cells seem like they have close to the same AUC as proximal tubules in the cortex. Can you add the y-axis to Figure 2D? Finally, in response figure 15, the number of intercalated cells appears too high as compared to PCs based on the images provided.

5. Inability to distinguish ascending and descending thin limb from TAL – you address this in the manuscript, but it is included here as a problematic cell group for completeness.

6. Myofibroblasts – The investigators suggest that the Cartana panel was too small to include markers to resolve myofibroblasts, but the panel appears to include Acta2 and PDGFRA (and PDGFRB). I do see that cells expressing Acta2 and PDGFRA co-cluster with Fibroblasts. However, the inability to distinguish these cells is not addressed in the manuscript. With acta2 and Pdgfra included, is this a different issue related to resolution or are these markers completely overlapping with fibroblast markers? In the raw data, there also appear to be some epithelial cells expressing these markers. Can myofibroblasts and fibroblasts be differentiated when using label transfer?

7. Neutrophils – their absence is not described in the manuscript, only in the reviewer response.

Minor critiques –

8. For the difference in PT cell types between figures, on line 323, it is stated “This allowed us to identify the same cell classes among the timepoints, including an injured PT state (injPT) that we have previously reported (Fig. 5A)”, but the opposite is stated in the response to reviewers: “For Figure 5, we annotated cell types from a combined dataset after correcting for batch effects using the Seurat integration algorithm. This leads to slight differences in the cell type labels. It is noteworthy that distinctions arise primarily in the PT: while single-sample clustering can discern different PT subtypes, this difference was not captured when clustering the combined dataset. This is likely because the PT is the primary cell type affected in IRI, and the transcriptional difference between injured and healthy PT is more pronounced than differences among PT subtypes. We have expanded our Methods section to provide a clearer explanation of our clustering methodology.” Thank you for expanding the methods, but the results still remain unclear. It would be helpful to inject some of this response to reviewers into the manuscript to explain discrepancies between Figures 2,3,4, and 5 if they cannot be resolved.

9. I cannot find that the merging of clusters is described in the manuscript as it is in the reviewer response.

REVIEWER COMMENTS

Reviewer #3 (Remarks to the Author):

In the manuscript “High resolution spatially resolved transcriptomic profiling of kidney injury and repair by RNA hybridization-based in situ sequencing”, the investigators resubmit a study using the Cartana platform to define cell type distribution in the murine and human kidney. Overall, the authors were responsive to several critiques, but there remain important clarifications needed. It is clear the authors care about providing high quality data and I will highlight the unaddressed critiques and remaining blindspots below to help them achieve this goal.

Main critique:

The authors employ a 200 gene panel, with most cell type definitions based on a single marker, yet only 13 cell types are identified, which is small number of cells identified. Some of these 13 cell types have unexpected distributions which remain unjustified. Depending on how cell types are counted, the authors appear to map more than 13 cell types, in some cases distinguishing PTS1, PTS2, and PTS3, as well as 6 immune subtypes (maybe you have 20 cell types?). After review of the raw datasets, there are even more markers which correspond to other cell types and it is unclear if the authors chose not to analyze these markers or if there were technical limitations of Cartana. One of 2 courses of action is suggested:

1. Re-run a subset of the samples with Cartana, optimizing probes for the problematic cell types to improve the confidence of and expand the number of cell types defined.
2. Present both the negative and positive data and address problematic cell types more clearly in the results. In the abstract, please clearly denote the number of cell types resolved on the Cartana platform and the high profile cells that could not be resolved. Add a section on limitations to the results detailing why important cell types failed.

There are several issues to clarify. While we utilized a 200 gene panel, of this total were 50 were kidney-specific genes. We selected these 50 based upon major cell type marker genes both in homeostasis and in kidney injury and repair from our prior mouse scRNA-seq datasets. We were clear about this in the introduction:

Lines 85-87: “We selected 50 custom kidney gene probes and added 150 pre-designed probes for immune and lung cell types.”

The reviewer feels that 13 cell types resolved is unexpectedly low, but this is this is a limitation of all imaging-based SrT techniques, rather than a quality issue in our study. For example, the first application of MERFISH (a similar imaging-based technique) in the brain included a 155 brain custom gene panel and yet they were able to detect only 9 cell types (<https://www.science.org/doi/10.1126/science.aau5324>).

Perhaps the reviewer expects SrT-based clustering to be similar to scRNA-seq based clustering. Indeed, in our previous scRNA-seq mouse IRI study with similar timepoints (Kirita et al, PNAS 2020), we were able to detect 26 distinct cell clusters. However, if we downsample that study to the same 200 genes we used in this study, the cell clusters are reduced from 26 to 11 (Figure 1):

Figure 1: Cell clustering on the scRNAseq data (Kirita et al, PNAS 2020). Cells were clustered using whole transcriptome (left panel), the 200 Cartana genes (middle and right panels) and colored by the original annotation (left and middle panel) or unannotated clusters (right panel).

We have previously called out this limitation of our Cartana dataset in our prior response letter, quoted here:

“With respect to the reviewer’s concern about a small number of cell clusters being detected in our dataset, we would emphasize that our Cartana data is capable of identifying nearly all major kidney cell classes, including those in the glomerulus (Podo and gEC), tubular epithelia (PT, TAL, DCT, PC, and IC), tubular interstitium (Fib and immune), urothelium (Uro), and endothelium (aEC). However, our Cartana data does not distinguish subpopulations as effectively as scRNA-seq. This limitation is, again, largely due to the constraint of including only 200 genes in a single run where we need to prioritize the selection of both cell-type specific and disease markers.”

Despite this limitation – lower numbers of cell clusters in our SrT dataset compared to scRNA-seq, our new cell segmentation and cell type classification pipeline is actually superior to the pipeline used by other groups for analyzing imaging-based datasets. For example, in the kidney MERFISH dataset published by Li et al (PMID: 36526371), they identify 9 cell types with many cell types containing cell doublets (such as EC-PT, KLH-EC and KCD-EC) (Fig. 2A in this response letter). We have now reanalyzed their dataset using the same pipeline that we developed and present in this manuscript, and we were able to identify 15 kidney cell types (Fig. 2B in this response letter). These include subpopulations of PT (such as PTS1/2 and PTS3), immune cells (such as Bcell and macrophages) and some rare cell types (such as gEC and urothelial cells). They all expressed their specific markers and have expected spatial distribution but were not detected in the original paper (Fig. 2C and 2D in this response letter). This provides further validation for the utility of our CellScopes pipeline that should be of broad general use to the spatial transcriptomics research community.

Figure 2: Reanalysis of a public mouse kidney MERFISH dataset (Liu et al, Life Sci Alliance 2022). **A.** Cell types identified from the original paper. **B.** Cell types identified from our analysis pipeline. **C.** Spatial distribution of the six new cell types that were not resolved from the original paper: PTS1/2, PTS3, Uro, macrophages, B cells and glomerular EC. **D.** Marker genes to define the six cell types.

1. Re-run a subset of the samples with Cartana, optimizing probes for the problematic cell types to improve the confidence of and expand the number of cell types defined.

We cannot add in new probes to the 200 gene panel post-hoc. This would require repeating the entire experiment with an entirely new set of probes. It is not possible to adjust a subset of probes because they are mixed together. Aside from doubling the costs for this already resource-intensive dataset and resource for the community, this would require ~12 months of work and analysis whereas the editors have requested our second revision to be submitted within four weeks.

2. Present both the negative and positive data and address problematic cell types more clearly in the results. In the abstract, please clearly denote the number of cell types resolved on the

Cartana platform and the high profile cells that could not be resolved. Add a section on limitations to the results detailing why important cell types failed.

We appreciate the reviewer's comments concerning the 'problematic' cell types mentioned in our study. In response, we have amended the abstract to explicitly include the number of cell types detected. We also show which lymphocyte genes were not detectable (see below). In addition, we have expanded the discussion to address why certain cell types were not identified in this study. It is important to clarify that the inability to detect some cell types is not a limitation of the dRNA HybISS technique itself. Rather, it stems from the constraints related to the number of genes included in our panel (we have emphasized this in our previous response letter). We disagree that "important cell types failed" and such a statement oversimplifies the limitations of the technology. We have added sections expanding on lymphocyte markers and neutrophils as requested by the reviewer (see detailed response below). Our results highlight that if researchers aim to investigate specific cell types or states such as neutrophils or TAL subtypes, it is important to incorporate additional markers for these cells in the original gene panel design. Such strategic enhancements in gene panel selection can significantly expand the scope and accuracy of cell type detection in spatially resolved transcriptomics studies. As requested, we now list the number of cell types that we resolved in the abstract. We also cite the specific number of genes from the custom kidney panel that were successfully detected (74%).

Problematic cell types:

1. Lymphocytes – The approach to lymphocytes is not clear in this study. Could the investigators add clarity to the following statement on line 456: "although we included 3 lymphocyte genes in our probe panel, we could not detect any lymphocytes in our dataset"? How were you able to map T cells from the scRNAseq data set in Figure 7D?

We have now carefully revised the relevant text in the discussion section to be more transparent about this limitation in the Discussion.

"A limitation inherent in the dRNA HybISS technology applied in this study is its restriction to a panel comprising 200 genes, which allowed for the inclusion of 50 custom genes in our probe design. Although our panel incorporated 11 genes specific to lymphocytes, most of these were undetected in our analysis. As a result, we were unable to identify heterogeneous lymphocyte subpopulations. Similarly, neutrophils were also not identified in our dataset, likely due to an insufficient number of neutrophil-specific markers in our gene panel design. This limitation has now been addressed in more recent commercial versions of this technology, Xenium, which now allows for 480 custom gene probes."

Regarding mapping T cells in Figure 7D, we were able to do this based on detection of Cd3e (low expressed but T cell-specific) and we gained confidence about this annotation based on high Pearson correlation of that cluster to a T cell cluster published by Conway et al., 2020 – Figure 7C of the current manuscript.

In the raw data on GEO, it appears that the following 11 lymphocyte genes are in the panel: Cd3e, cd3d, cd3g, cd8a, cd8b1, cd19, cd4, cd79a, cd79b, foxp3, ly6e. This is not counting JCHAIN or Igkc for plasma cells or NKG7 for NK cells or quite a few others (interleukins, NCAM1, etc). Thus, there are more than 3 lymphocyte genes. Did these probe sets fail or were they crowded out by more highly expressed genes? Basically, please include the data on your experience with all these immune cell markers in a limitations results section so readers of Nature Communications know how these classic markers of immune cells perform on the Cartana platform.

We now provide a new figure to show the average expression of the lymphocyte markers that the reviewer mentions (Fig. 3 in this response letter). Most of the 11 lymphocyte markers were not detected. We also did not detect the NK cell marker gene NKG7. For the markers that were successfully detected (such as Ly6e and Igkc), we included them in Figure 7 to define the immune subtypes. As requested by the reviewer and for transparency, we now show this negative data in Supplementary Figure S18C. We would like to point out that it is unclear to us (as non-immunologists), whether we would in fact expect all 11 lymphocyte marker genes to be expressed in a kidney injury-repair timecourse, so it is possible that some of these are true negative results.

Figure 3: Expression of the lymphocyte and neutrophil markers in the immune subpopulations at week 6.

2. Vascular Smooth muscle cells – TAGLN appears to be included in your panel. Why is this excluded from the study? The raw data suggests it confidently mapped to certain cells in a separate cluster to that of the fibroblasts.

We did not exclude Tagln from the study. All genes are included in the analysis and contributed to the final cell clustering. Since the reviewer is interested in Tagln (and in comment #6 below), we now provide new figure showing fibroblast subclustering on Day 2 (Fig. 7 in this response letter). This revealed two subpopulations of fibroblasts; pericyte/myofibroblasts and fibroblasts. Tagln is primarily expressed in the pericyte/myofibroblast cluster (Fig. 7D in this response letter). This expression pattern is consistent with the expression in our previous snRNA-seq data (Kirita et al 2020) which can also be viewed on our online data visualizer.

3. Proximal tubule – This study maps a different number of cell types in figure 2 (N=13), figure 3 (N=14), Figure 4B (N=15), Figure 4D (N=14), and Figure 5 (N=13). Sometimes, the PT-S1, S2, and S3 are distinguished, and sometimes the injured PT is removed. Then in Figure 7, the FR-PT is introduced which is not included in these prior figures. To help the reader, is it possible to provide 2 common sets of cell types, one based on unbiased Cartana clusters from merged/normalized samples and a second based on a common set of scRNA-seq label transfer clusters throughout for all specimens? If not, please justify why a different number of clusters is required. What genes define the FR-PT cell type? How did you include the slc5a12 probe set?

First, the reviewer correctly points out an inadvertent error that we made in the annotation differences between Figure 4B and 4D. We inadvertently used the cell annotation from the

integrated data in Figure 5A for Figure 4B. We have now corrected this for consistency (please see updated Figure 4 in the main manuscript). We appreciate this point.

Regarding the request to “provide 2 common sets of cell types, one based on unbiased Cartana clusters from merged/normalized samples and a second based on a common set of scRNA-seq label transfer clusters throughout for all specimens,” we think this will actually confuse the reader, not clarify. The reason there are different numbers of cell clusters in these figures results entirely from different PT cell types/states. For example, Figure 2 shows day 2 after IRI. At this timepoint, there is an injured PT (injPT) cluster that is not present in healthy kidney (nor is expected to be present). In Figure 3, we compare male and female healthy kidney – so the injPT cluster is absent. However, in healthy kidney we can distinguish PT S1, S2 and S3 – for a net gain of one cell cluster as noted by the reviewer. In Figure 2, we cannot distinguish PT S1, S2 and S3 because injury causes downregulation of PT segment-defining markers (Fig. 4 in this response letter and Supplementary Figure S15) – leaving us with one single “HealthyPT” cluster instead of three separate healthy S1, S2 and S3 clusters. Essentially, we can only distinguish S1 vs. S2 vs. S3 in the healthy sample but not the four post-injury samples. We now clarify our reasoning more precisely in the results to explain the differing number of clusters.

Figure 4: Expression of PT subpopulation genes across timecourse.

Regarding FR-PT, this is a good point. We have now provided a supplementary figure showing independent clustering of the week 6 sample revealing FR-PT at this time point (Fig. 5A in this response letter and Supplementary Figure S18 in the revised manuscript). The FR-PT defining gene that we used is *Vcam1* which is uniquely expressed by this cluster and is consistent with our previous snRNA-seq dataset that defined the FR-PT state (Kirita et al, 2020). Regarding *Slc5a12*, we are not sure what the reviewer is asking, but we now add its expression to new Figure S17 and below in 5B.

Figure 5: Cell types defined at the week 6 sample. **A.** Three PT states were classified at week 6: Healthy, injured and failed repair PT. **B.** Expression of the PT disease state specific markers.

4. Intercalated cells – The intercalated cell distribution in this study is unexpected. In response Fig 13A, I see the interposed ICs with the PCs. However, the distribution of intercalated cells in the new Supp Fig 2-5 is abnormal - why are there multiple tubules composed almost entirely of intercalated cells? These are seen in the Upper third of the image in Supp Fig 2D/G, 3C/E, 4C/E and lower third in Supp Fig 5C (this one is upside down). Have these tubules been miscategorized? There are few if any principle cells colocalizing in these yellow tubules.

We agree with the reviewer that response Figure 13A (from our Cartana dataset) shows interposed IC's with PCs. We want to re-emphasize that Supp Figure 2-5 is not from our Cartana dataset – this is publicly available data from the 10X website. We used it to demonstrate the functionality of CellScopes in comparison with other tools such as Seurat, Scanpy, and Giotto. The reviewer makes an interesting observation regarding the spatial arrangement of IC and PC in the human kidney cortex, but this is not our dataset so it is difficult to speculate on the reason for this finding. What we have now done in response to this comment is reanalyze a human kidney Visium dataset (also downloaded from the 10x Genomics website), which shows consistently that IC and PC gene expression actually does not generally overlap in the same spot (Figure 6 below in this response letter). This may reflect that in human cortex the relationship between IC and PC may be different than in mouse, or in papilla. In other words, based on these results we cannot say for sure if this expression pattern is 'unexpected.' We do feel that this is beyond the focus of our manuscript analyzing kidney injury and repair.

Figure 6: Reanalysis of a human kidney Visium dataset from 10x Genomics. Two principal cell markers (AQP2 and FXYD4) and two intercalated cell markers (SLC26A7 and ATP6V0D2) were shown.

Figure 2D still seems incorrect when comparing between rows. The proportion of ICs, JGA, and other lowly represented cells seem like they have close to the same AUC as proximal tubules in the cortex. Can you add the y-axis to Figure 2D?

In Figure 2D, we present a stacked density plot to visualize the distribution of various cell types and transcripts across kidney depth. This is not intended for direct comparison of absolute quantities of each cell type or transcript compared to others. Density plots are fundamentally designed to provide insights into the distribution and density of data points, rather than their absolute counts. In such plots, the y-axis represents the probability density of the data, not the actual count or percentage of cells or transcripts. Thus, the height at any given point on the plot indicates the density of data points (cells or transcripts in our case) within that region, not the total number. This aspect makes the specific values on the y-axis less important for our analysis. Therefore, we have utilized the y-axis space to prominently display cell and gene names. That being said, in Figure 5A we do show comparison of cell proportions within samples as well as between all timepoints across the full IRI timecourse to address this comment.

Finally, in response figure 15, the number of intercalated cells appears too high as compared to PCs based on the images provided.

This is because the cell type annotations in Figure 15 are based on data from Visium. As the reviewer will be aware, in Visium each spot is approximately 55 microns in diameter. This size significantly exceeds the scale of single-cell resolution, which inherently affects the precision of cell type annotation. Each spot in this figure is labeled with a cell type as a conventional practice in Visium analysis adopted by the whole community. However, Visium lacks single cell resolution, these annotations cannot be relied upon for precise assessment of cell type proportions. For a more accurate representation of cell type proportions within the kidney, we recommend that the reviewer refer to Figure 5A. This figure provides a clearer and more precise assessment of cell type distribution, thanks to the higher resolution nature of our Cartana data.

5. Inability to distinguish ascending and descending thin limb from TAL – you address this in the manuscript, but it is included here as a problematic cell group for completeness.

Thank the reviewer for highlighting the inability to identify the subpopulations of TAL once again. We have acknowledged this limitation in the earlier revision (Lines 218 – 223). Quoted here: “However, due to the constraint on the number of genes included, cell types lacking marker genes in our probe panel were undetectable in our data. These include the parietal epithelial cells (PEC), the thin limb of the loop of Henle (ATL-DTL), and certain subpopulations within a cell type (e.g., mTAL, cTAL1, and cTAL2 from Kirita et al dataset17 were indistinguishable within the TAL cell cluster) (Supplementary Fig. 8).”

6. Myofibroblasts – The investigators suggest that the Cartana panel was too small to include markers to resolve myofibroblasts, but the panel appears to include Acta2 and PDGFRA (and PDGFRB). I do see that cells expressing Acta2 and PDGFRA co-cluster with Fibroblasts. However, the inability to distinguish these cells is not addressed in the manuscript. With acta2 and Pdgfra included, is this a different issue related to resolution or are these markers completely overlapping with fibroblast markers? In the raw data, there also appear to be some epithelial cells expressing these markers. Can myofibroblasts and fibroblasts be differentiated when using label transfer?

We thank the reviewer for this point. While we do not see a myofibroblast cluster from global clustering as shown in the manuscript, we have now performed additional analysis to subcluster the fibroblasts on day 2, the time point where fibroblasts are most abundant. We have added a supplementary figure (Supplementary Figure S10B-E) to show that our Cartana data can also distinguish a pericyte/myofibroblast cluster and fibroblast cluster (Fig. 7 in this response letter).

Figure 7: Subclustering the fibroblasts from the Day 2 IRI sample. **A.** UMAP to show the fibroblast subtypes. **B.** Spatial distribution of the pericyte in the day 2 kidney. **C.** Spatial distribution of the fibroblast. **D.** Expression of the pericyte (Tagln and Acta2) and fibroblast genes (Col1a1 and Col1a2).

7. Neutrophils – their absence is not described in the manuscript, only in the reviewer response.

We have now described this limitation in the discussion of the revised manuscript. We assume that our reviewer responses will be published as well which is why we did not highlight this previously.

Minor critiques –

8. For the difference in PT cell types between figures, on line 323, it is stated “This allowed us to identify the same cell classes among the timepoints, including an injured PT state (injPT) that we have previously reported (Fig. 5A)”, but the opposite is stated in the response to reviewers: “For Figure 5, we annotated cell types from a combined dataset after correcting for batch effects using the Seurat integration algorithm. This leads to slight differences in the cell type labels. It is noteworthy that distinctions arise primarily in the PT: while single-sample clustering can discern different PT subtypes, this difference was not captured when clustering the combined dataset. This is likely because the PT is the primary cell type affected in IRI, and the transcriptional difference between injured and healthy PT is more pronounced than differences among PT subtypes. We have expanded our Methods section to provide a clearer explanation of our clustering methodology.” Thank you for expanding the methods, but the results still remain unclear. It would be helpful to inject some of this response to reviewers into the manuscript to explain discrepancies between Figures 2,3,4, and 5 if they cannot be resolved.

We have expanded the text to more directly address this in response to comment 3 above.

9. I cannot find that the merging of clusters is described in the manuscript as it is in the reviewer response.

Thank you for this comment. We have now revised the text in the manuscript to make it consistent with the text in the previous response letter.

REVIEWERS' COMMENTS

Reviewer #3 (Remarks to the Author):

The authors were sufficiently responsive. This study will likely prove valuable to the scientific community as they weigh the use of the Cartana (and its derivative Xenium) platform. I wish the investigators good fortune in their future work.

Reviewer #3 (Remarks on code availability):

I reviewed the code. Generally, the tutorials and code seem extensive and helpful.

In my initial review, I identified select circumstances where the code was not available to create a particular object from the raw data, and instead, the authors provide only endpoint code to display the figure. These circumstances seem to be rectified, but I did not review this as thoroughly on the revision.